# Hi-C metagenome sequencing reveals soil phage–host interactions

Ruonan Wu [1], Michelle R. Davison[1], William C. Nelson [1], Montana L. Smith [1], Mary S. Lipton[1], Janet K. Jansson [1], Ryan S. McClure[1], Jason E. McDermott [1,2] & Kirsten S. Hofmockel [1,3] ✉

Bacteriophages are abundant in soils. However, the majority are uncharacterized, and their hosts are unknown. Here, we apply high-throughput chromosome conformation capture (Hi–C) to directly capture phage-host relationships. Some hosts have high centralities in bacterial community co-occurrence networks, suggesting phage infections have an important impact on the soil bacterial community interactions. We observe increased average viral copies per host (VPH) and decreased viral transcriptional activity following a two-week soil-drying incubation, indicating an increase in lysogenic infections. Soil drying also alters the observed phage host range. A significant negative correlation between VPH and host abundance prior to drying indicates more lytic infections result in more host death and inversely influence host abundance. This study provides empirical evidence of phage-mediated bacterial population dynamics in soil by directly capturing specific phage-host interactions.

Viruses are highly abundant in soil, with estimates of $10^7$ to $10^{10}$ viral-like particles per gram of soil[1], and influence the regulation of host population dynamics[2]. Current studies suggest that drier soils favor a temperate (lysogenic) lifestyle for bacteriophages (or phages), in which they reside as prophages in their bacterial hosts[3–5]. By contrast, under wet soil conditions, phages tend to become lytic, actively replicating, and killing their hosts[3–5]. Transitions between lysogenic and lytic lifestyles can have significant impacts on host abundances and thus soil microbial composition and function. However, it remains challenging to identify which phage infects which host(s) when examining the complex microbial community in soil. This knowledge is needed for a more accurate understanding of the impact of soil phages on the soil microbial community composition and function and how shifts in soil moisture impact specific phage–host interactions.

Most current techniques for identifying phage-host pairs rely on indirect sequence-based evidence. These methods include matching CRISPR spacers to phage genomes[6,7] and searching for similarities between phage and host genomes using alignment-dependent (e.g., VPF-Class[8]) or alignment-free methods (e.g., WIsH[9], VHM[10], and PHP[11]).

These approaches have provided insights into which host taxa are potentially infected by soil phages. However, phage infection is a dynamic process. For example, changing environmental conditions over time could cause changes in the susceptibility of host microbes to phage infection. Meanwhile, phage genomes constantly undergo mutations and recombination[12]. Therefore, host predictions based on microbial genomic features could fail to capture viral infections at the time of sampling. Misinterpretation of phage–host interactions may lead to a biased perspective of phage impacts on the soil microbiome.

In this work, we overcome the current limitations of capturing phage–host interactions at the time of sampling by using high-throughput chromosome conformation capture (Hi–C) metagenomic sequencing. The Hi–C technique relies on chemically cross-linking the DNA of an infecting phage (phage that enters the intact host cell) to the genome of the host organism. This approach has previously been used to track phage infections in the human gut[13,14]. No previous study, to our knowledge, has applied Hi-C sequencing to soil samples to directly capture specific phage-host interactions at the time of sampling. We apply the Hi–C sequencing approach to grassland soils that were

[1]Earth and Biological Sciences Directorate, Pacific Northwest National Laboratory, Richland, WA, USA. [2]Department of Molecular Microbiology and Immunology, Oregon Health & Science University, Portland, OR, USA. [3]Department of Agronomy, Iowa State University, Ames, IA, USA. ✉ e-mail: kirsten.hofmockel@pnnl.gov

collected either before or after drying during a two-week incubation. Desiccated soil was detected with potentially lysogenic phages that have a broader host range and/or target hosts with fitness advantages. This study provides knowledge about how phage–host interactions change as soil moisture shifts and has implications for predicting the consequences of climate change on soil ecology.

## Results

### Identification of infections in the soil microbiome using Hi–C metagenome sequencing

Shotgun metagenomes sequenced from each of the replicate soil samples (Supplementary Data 1a) were assembled and screened for viral sequences as shown in the workflow diagram (Fig. 1). A total of 583 viral contigs were identified (Supplementary Data 1b) and clustered into 479 viral operational taxonomic units (vOTUs, Supplementary Data 2 and Supplementary Data 3). The sequences all corresponded to bacteriophages and therefore we refer to the viruses as phages. Nearly half of the vOTUs could not be classified, with the rest assigned to the class Caudoviricetes. A viral tree composed of three major clades was used to visualize the genome-wide similarities of the vOTUs within each sample (Fig. 2).

To enable the targeted detection of host-associated phages, we applied Hi-C metagenomics to the same soil samples (Supplementary Data 1c). Specific phage–host interactions were captured using the Hi–C approach through chemical cross-linking of the phage and host DNA molecules that were co-localized within the same cell at the time

of sampling (Fig. 1). Phage–host linkages that were identified in the Hi–C metagenomes were quality-filtered (Supplementary Data 1d and Supplementary Data 4) to identify 118 unique phage-host pairs (Supplementary Data 5). A total of 148 unique metagenome-assembled genomes (MAGs) spanning nine bacterial phyla were binned by following the Hi-C metagenome deconvolution protocol (details in Methods; Supplementary Data 6). Phages belonging to 19 of the 479 detected vOTUs were assigned to their respective hosts represented by unique bacterial MAGs via the identified phage–host pairs (Supplementary Data 5, orange and dark blue cells in the rings of Fig. 2). The host-associated phages accounted for 5.3% to 15.0% of the total phage sequence abundance detected in the samples (Supplementary Data 3).

To demonstrate that Hi-C sequencing can capture viral infections at the time of sampling, we compared the phage–host links detected by Hi–C with those predicted by CRISPR spacer matching, which is currently the main bioinformatic method for viral host prediction[15,16]. A total of 124 CRISPR spacers recalled from the CRISPR arrays in MAGs were matched to phage contigs, generating 121 unique phage-host links (Supplementary Data 7). Although the number of phage–host links predicted by the CRISPR spacer method and detected by the Hi–C method are comparable, none of the Hi–C links were detected using the CRISPR spacer approach (Supplementary Data 5 and Supplementary Data 7). Because the CRISPR-Cas system provides an adaptive immunity to host cells, the immunity memory based on prior viral infections[17] may not detect more recent or current viral infections.

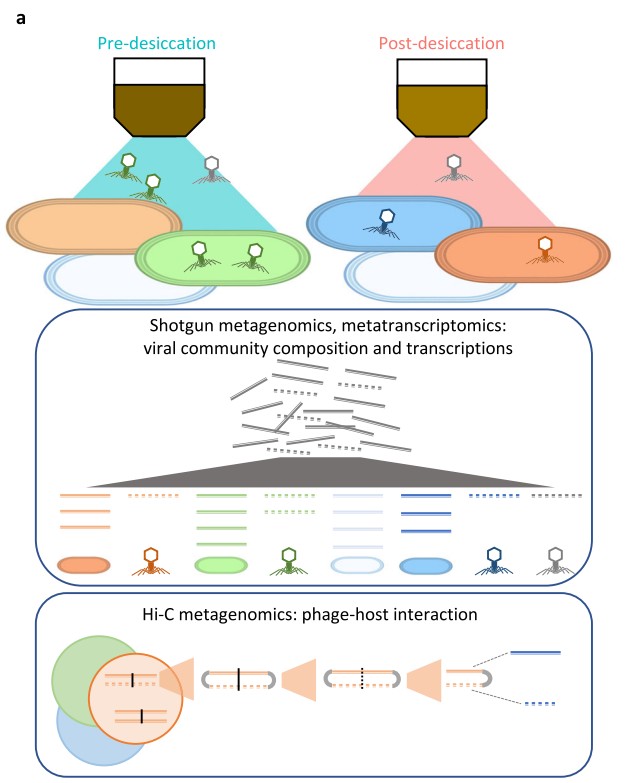

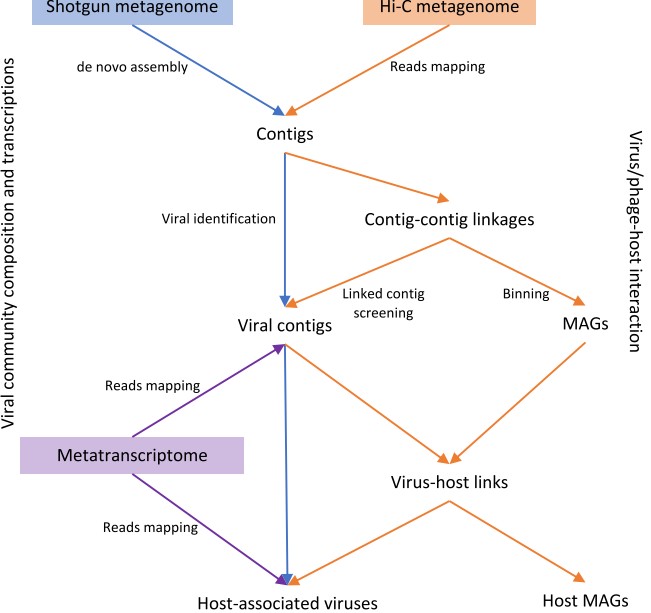

**Fig. 1 | Schematic of the experimental design and data analysis workflow. a** Description of sample treatment and the collected sequencing data. Soil samples were collected pre- (75% water holding capacity, represented by the first soil jar icon) and post-desiccation (soil dried to consistent weight, represented by the second soil jar icon). Shotgun metagenomes (sequenced on the total DNA, demonstrated in the first box), (bulk) metatranscriptomes (sequenced on the total RNA, demonstrated in the first box), and Hi-C metagenomes (sequenced on the cross-linked DNA, demonstrated in the second box) were generated from each replicate sample. **b** Data analysis and integration. Shotgun metagenome-assembled contigs were used to screen for viral contigs. The normalized read coverages of the identified viral contigs in shotgun metagenomes and metatranscriptomes were used to detect the viral (phage) community compositions and transcriptional activities in the soil samples before and after desiccation. Hi-C metagenomes containing the sequenced cross-linked DNA of the extracted host microbial cells were used to identify contig-contig linkages. The quality-filtered linkages were used to cluster linked contigs into metagenome assembled genomes (MAGs) and to identify phage-host pairs. Co-analysis of the paired shotgun and Hi-C metagenomes was performed for the detection of phage-host interactions under wet and dry soil conditions and to determine phage-host interactions.

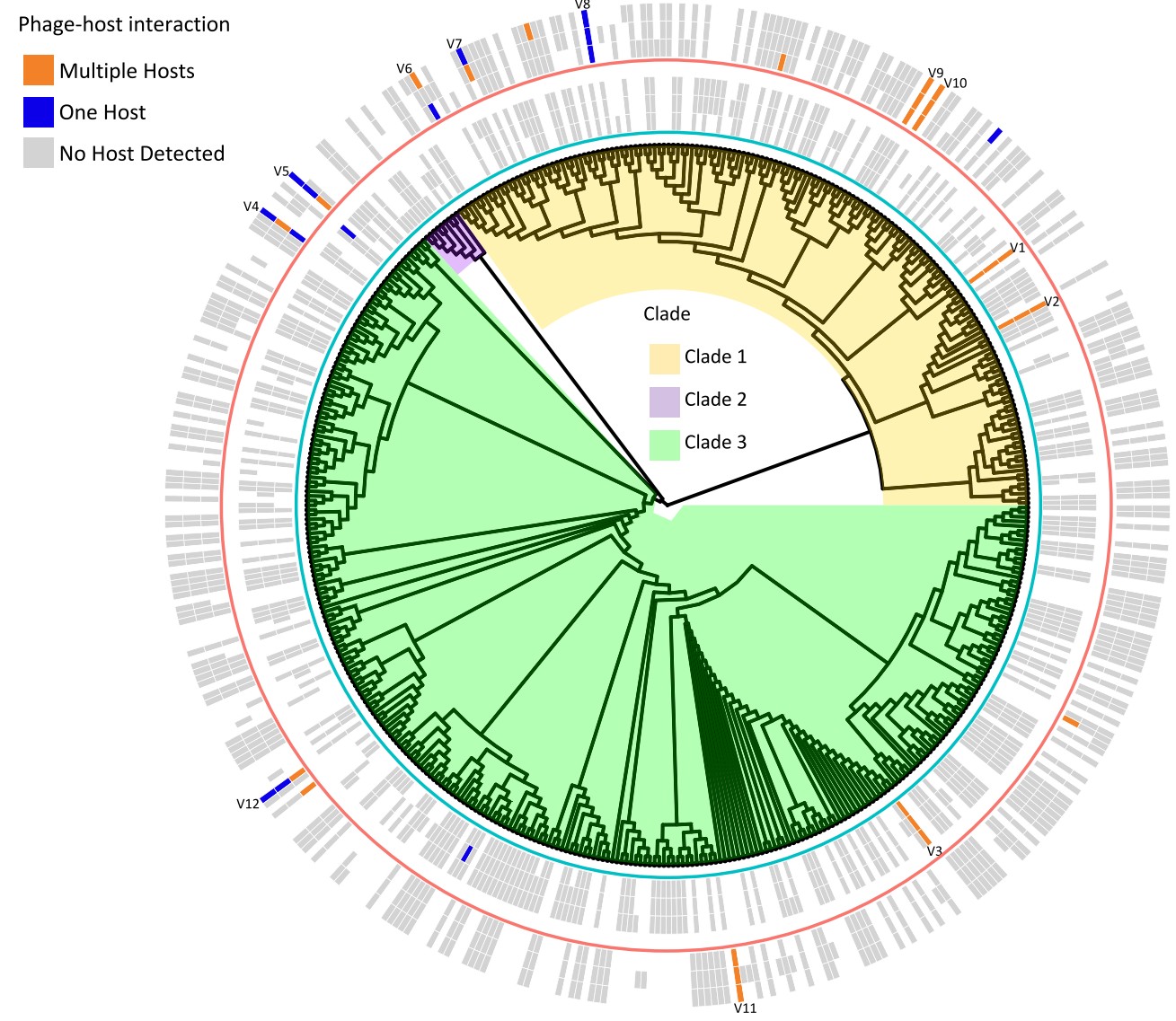

**Fig. 2 | Soil phage−host interactions revealed using Hi−C metagenomics.** Hi−C metagenomics was used to characterize host-associated vOTUs in pre- and post-desiccation soils. The tree of all the detected vOTUs based on genome-wide similarities is shown in the center with the three major clades colored in yellow (Clade 1), purple (Clade 2) and green (Clade 3). The two heatmaps outside the tree indicate the vOTUs that were detected in each of the three replicate samples. The outlines underneath each heatmap indicate the experimental conditions: Pre-desiccation (inner circle, blue); Post-desiccation (outer circle, red). The cells of the heatmaps are colored by the type of phage-host interaction of each vOTU: Multiple Hosts in orange, One Host in dark blue, and No Host Detected in gray. Empty (white) cells indicate the vOTUs were not detected in the sample. The vOTUs linked to the same host(s) in at least two replicates are labeled. Source data are provided as a Source Data file.

Hi−C sequencing overcomes this challenge by directly capturing phage−host interactions at the time of sampling.

## Hi−C metagenomes and metatranscriptomes reveal increased lysogeny following soil drying

The Hi−C metagenome sequencing approach was combined with shotgun metagenome and metatranscriptome sequencing to investigate how soil drying impacts phage−host interactions. Results were compared from before and after a two-week drying incubation (pre- and post-desiccation, Fig. 1) that simulated summer desiccation that frequently occurs in these arid grassland soils. Screening of the shotgun metagenomes revealed that the soil phage community shifted in response to soil drying. Only 18.0% of the total vOTUs were detected in both the pre- and post-desiccation soils (Fig. 2). Similar to the phage communities detected in the shotgun metagenomes, the host-associated phage communities in the Hi−C metagenomes were

significantly impacted by soil drying ($p <= 0.005$, Fig. S1). A significantly larger fraction of detected vOTUs (relative richness) were host-associated in post-desiccation soil when compared to pre-desiccation soil ($p < 0.05$, Fig. 3a). However, the relative abundance of the host-associated vOTUs (versus the total abundance of all recovered vOTUs) was not significantly different between pre- and post-desiccation soils ($p = 0.18$, Fig. 3b).

We further applied complementary metatranscriptomes to assess the transcriptional profiles of the host-associated phage (Supplementary Data 1e, Fig. 1b). Metatranscriptomic data revealed that a higher percentage of transcriptionally active vOTUs were identified as host-associated following soil desiccation (Fig. 3c). However, the fraction of transcripts which mapped to those host-associated vOTUs was lower post soil desiccation (Fig. 3d), suggesting that the average transcriptional activity of the host-associated vOTUs declined following soil drying.

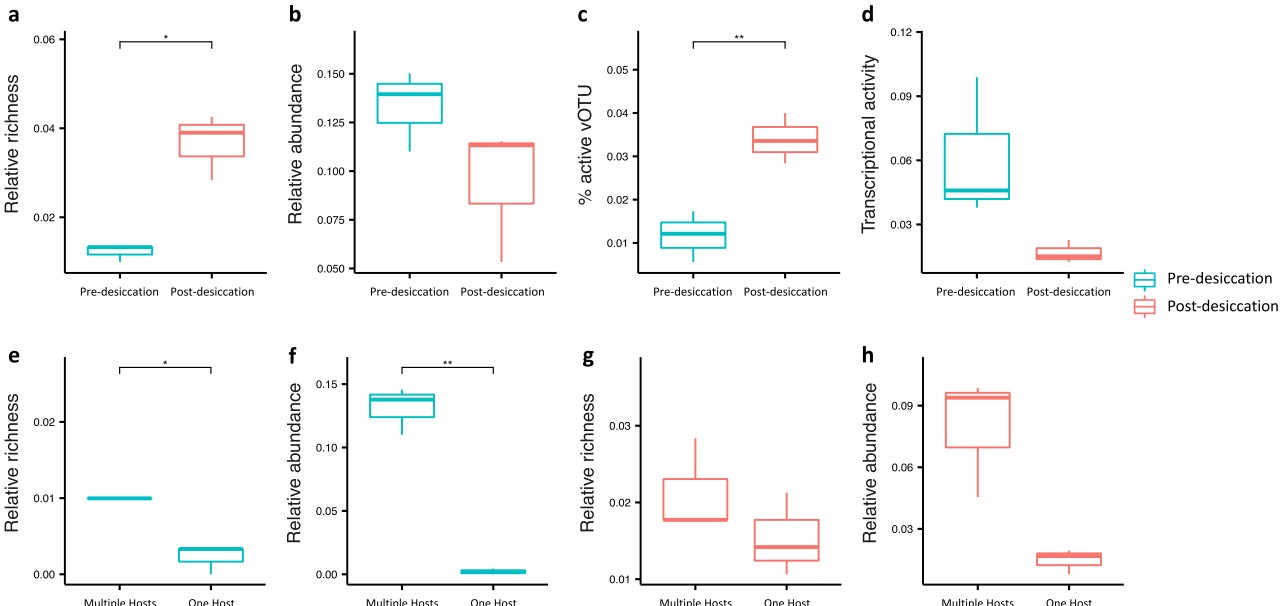

**Fig. 3 | Richness, abundance, and transcriptional activities of host-associated vOTUs. a** Relative richness of host-associated vOTUs to total vOTU richness in pre- and post-desiccation soil incubations. The relative richness of the host-associated vOTUs was calculated by dividing the number of host-associated vOTUs by the number of the total recovered vOTUs. **b** Relative abundance of the host-associated vOTUs to total vOTU abundance. **c** Percentage of transcriptionally active vOTUs that were host-associated. **d** Percentage of transcripts that were mapped to the host-associated vOTUs. Comparison of relative richness **e** and relative abundances **f** of the vOTUs infecting multiple hosts (Multiple Hosts) and the vOTUs infecting one host (One Host) in wet soils. Comparison of relative richness **g** and relative abundances **h** of the vOTUs infecting multiple hosts (Multiple Hosts) and the vOTUs infecting one host (One Host) in post-desiccation soils. The relative richness of the vOTUs infecting multiple hosts or one host was calculated by dividing the number of those vOTUs by the number of the total recovered vOTUs. Each panel contains a box plot for comparing the three biological replicates with the pre-desiccation treatment ($n = 3$) against the three biological replicates with the post-desiccation treatment ($n = 3$). Boxes colored in blue and red represent Pre-desiccation and Post-desiccation soil incubation treatments, respectively. In each boxplot, the top and bottom of each box represent the 25th and 75th percentiles, and the center line indicates the median. The upper and lower whiskers of each box represent the maximum and minimum values detected in the three biological replicates, respectively. The differences between the pre-and post-desiccation treatments were assessed using a two-sided t-test. The significant differences in all panels are highlighted by asterisks, with * representing $p < 0.05$ and ** $p < 0.01$. The exact p-values of the comparisons shown in panels **a**–**h** are 0.02, 0.18, 0.009, 0.06, 0.02, 0.006, 0.28 and 0.06, respectively. Source data are provided as a Source Data file.

## Hi-C reconstructs soil phage-host infection network

MAGs identified as phage hosts by Hi-C sequencing included members of Acidobacteria, Actinobacteria, Chlamydiae, Gemmatimonadota, and Proteobacteria (Supplementary Data 5 and Supplementary Data 6). A single MAG linked to one or more phages by Hi-C sequencing represents one host population. Two of the five host-associated vOTUs in pre-desiccation soils were linked to a single host MAG, and each of these linkages was only observed in one of the three replicates (Fig. 2). The other three vOTUs were linked to multiple hosts (V1, V2 and V3), and these linkages were consistently found in all three replicates (Fig. 2).

An infection network containing phage-host pairs that were detected in at least two replicate samples was constructed and showed that the same hosts for V1, V2 and V3 were observed in multiple replicates (Fig. 4a). V1 phages infected members of both Acidobacteria and Alphaproteobacteria, while V2 phages infected bacteria belonging to Rhizobiales/Hyphomicrobiales (Rhizobiales has been updated to Hyphomicrobiales in the current NCBI taxonomy[18]) and two MAGs with unclassified taxonomy. All four hosts of V3 phages remain unclassified. A larger number of phage-host pairs were found in the post-desiccation soils, with a total of 14 vOTUs having linkages to hosts (Fig. 2). In most cases, the association between vOTUs and hosts was consistent across the replicates. For example, V8 phages were found to infect the same Actinobacterial host in all replicates (Fig. 4a). V9, V10 and V11 phages were all associated with multiple hosts. The vOTUs that were linked to multiple hosts had both higher richness and abundances (relative to the total recovered vOTUs) than those with only one host detected

(Fig. 3e versus Fig. 3f, Fig. 3g versus Fig. 3h). This was especially true in the pre-desiccation soils ($p < 0.05$, Fig. 3e versus Fig. 3f) where phages were also generally more transcriptionally active (Fig. 3d).

The phage-host pairs included in the infection network were exclusively found in either pre- or post-desiccation soil (Fig. 4a), indicating that the soil drying process altered phage-host interactions and infections. Only one host population, with the representative MAG classified as Alphaproteobacteria (B94), was infected by phages both pre- (V1 and V2) and post-desiccation (V5 and V11). In post-desiccation soils, a host population represented by Actinobacterial MAG B117 was targeted by six vOTUs (V4, V7, V8, V9, V10, and V12). vOTUs in post-desiccation soils were generally associated with a relatively lower richness of host MAGs compared to pre-desiccation soil (Fig. 4a).

## Phage hosts are central community members

Community co-occurrence networks were used to identify microbial members that have high centrality, which is to say that they are connected to many other species (high degree) or occupy key bridge points in the network (high betweenness). Centrality has been recognized as a strong proxy for the importance of a species to a biological system[19]. We, therefore, cross-referenced bacterial community co-occurrence networks with the infection network to gauge to what degree soil phages target highly central members of the bacterial community network. Co-occurrence networks were constructed based on transcript abundance of the arginine-tRNA ligase (*argS*) housekeeping gene as a proxy for MAG abundance (Supplementary Data 8 and Supplementary Data 9).

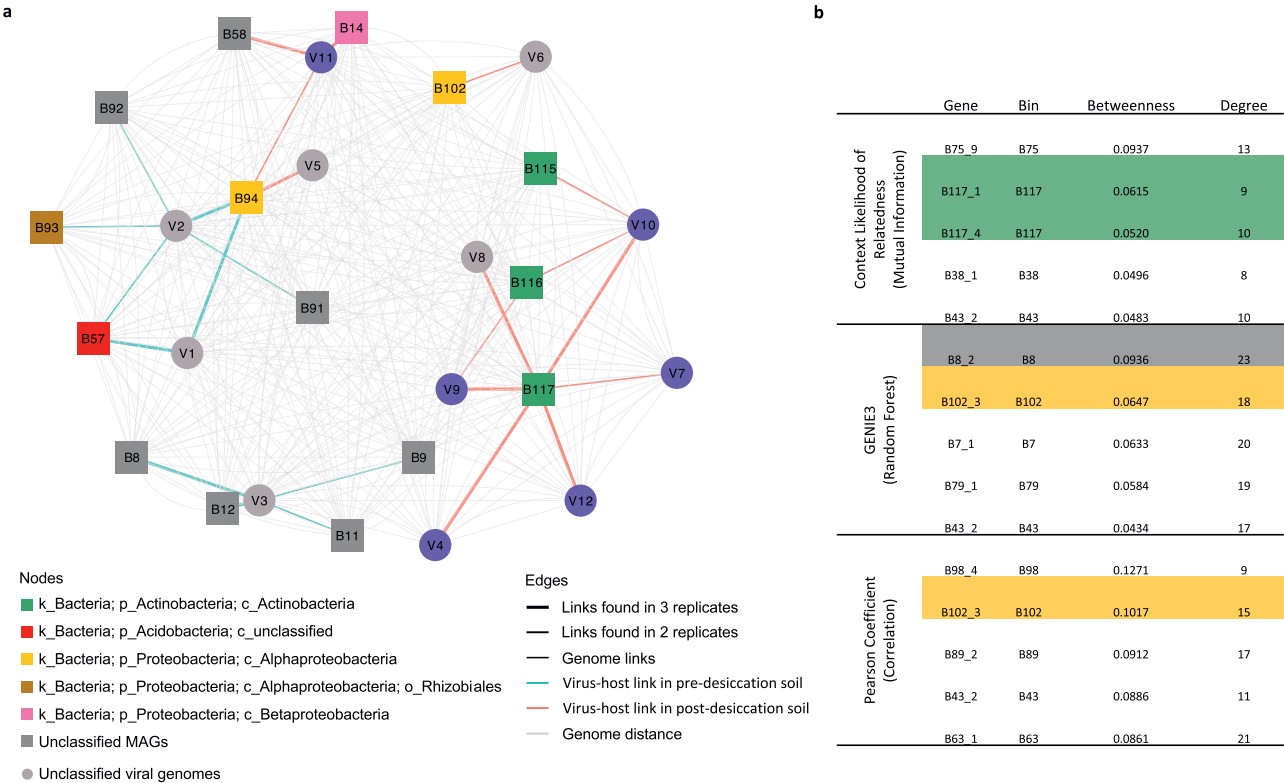

**Fig. 4 | Phage–host infection network and community co-occurrence analysis.**
**a** Phage–host infection network showing phage-host pairs that were detected in at least two replicate soils. vOTUs and their linked hosts are shown as nodes in circular and rectangular shapes, respectively. The nodes are colored by taxonomic assignment. Host–host and vOTU-vOTU edges are colored in gray and the vOTU-host edges are colored by treatment (Pre-desiccation: blue; Post-desiccation: red). Edge thickness corresponds to the number of replicates in which the vOTU-host interaction was observed. **b** The top five metagenome assembled genomes (MAGs) by betweenness centrality using three network inference methods. The MAGs identified as phage hosts are highlighted and colored by their taxonomic assignment following the same color scheme used in panel **a**. Source data are provided as a Source Data file.

Three complementary network inference methods, context likelihood relatedness (CLR), random forest (GENIE3) and Pearson correlation coefficient, were used to construct community co-occurrence networks[20,21]. Together these approaches provided a more comprehensive identification of bacterial taxa that were central to the community structure. The resulting networks contained 33 to 64 MAGs (Supplementary Data 8). MAGs identified as phage hosts were some of the most central nodes in the networks. For example, MAG B117, representing an Actinobacterial population, was associated with several vOTUs that were detected in post-desiccation soil (Fig. 4a). The two B117 *argS* genes were the second and third most central nodes in a network inferred using CLR (out of 64 nodes in total, Supplementary Data 8) (Fig. 4b). This centrality ranking was based on betweenness centrality, a measure of the extent to which a node (MAG) serves as a connection point between other nodes. When looking at degree (the number of connections (edges) a node has), these two B117 *argS* nodes were the second and fourth most central nodes. Host MAGs B8 and B102 were the first and second most central nodes as determined by betweenness and the first and fourth as determined by degree in a GENIE3 network (Fig. 4b). MAG B102 was also found to be highly central in the Pearson correlation coefficient network (Fig. 4b). We also confirmed that centrality in the network was not merely a function of a MAG being highly abundant (Table S1). Some MAGs that were central were abundant, but many were not, showing that abundance is not the main driver of centrality in our host co-abundance network.

### Phage infection regulates host population dynamics

Alphaproteobacteria, Chlamydiae, Acidobacteria and some unclassified phyla were the major host taxa in pre-desiccation soil, and Alphaproteobacteria, Betaproteobacteria and Actinobacteria were the major host taxa in post-desiccation soil. We calculated the average viral copies per host (VPH) as an estimation of the degree of phage infection across host populations. The average VPH and the VPHs of some host populations (e.g., Actinobacteria) were significantly higher after soil drying ($p < 0.05$, Fig. 5a). A higher average VPH indicates that either a higher fraction of the host populations are infected by phages (more individuals infected) or a subset of the host populations are infected by a higher number of phages (multiple phages per individual). As such, comparing VPH alone in soils pre- and post-desiccation can indicate a change in phage-host relationships but cannot determine the nature of the change. Because we observed lower transcriptional activity and lower relative abundance for the phages infecting multiple hosts in post-desiccation soil, the higher VPH after soil drying may suggest that phage infection was more prevalent among the individuals within the host population (more individuals infected). To further investigate the potential impact of phage infection on host population dynamics, we fit regression models to test the relationship between VPH and host abundance. Prior to soil desiccation, VPH and host abundances were significantly negatively correlated ($p < 0.001$, Slope = -0.41, Fig. 5b). This result may suggest phage infection in pre-desiccation soil inversely influenced host abundance, indicative of lytic infections. No obvious relationship was observed in post-desiccation soil ($p = 0.59$, Slope = 0.089, Fig. 5c).

## Discussion

This is a pioneer study to provide empirical evidence of phage-host associations in soil by applying Hi-C metagenomics to chemically link phages to their infected hosts. Previous attempts to identify

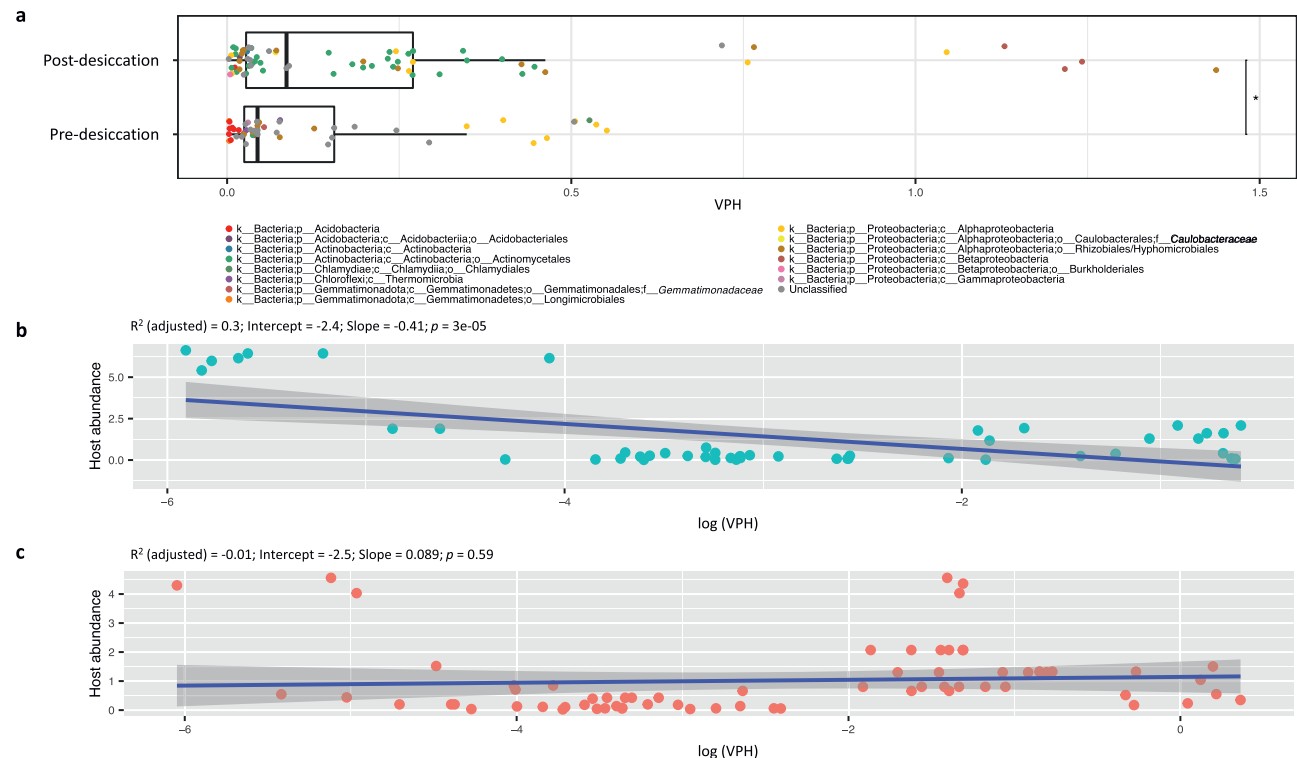

**Fig. 5 | Prevalence of phage infections in bacterial host populations. a** The viral (phage) copies per host (VPH) detected in soils pre- and post-desiccation. Each data point is a single de-replicated MAG linked to phage by Hi-C sequencing and represents a phage host population. The phage host populations are identified from at least two of the three biological replicates of the soils with the pre-desiccation ($n = 49$ unique populations) or post-desiccation treatment ($n = 69$ unique populations). Datapoints are colored by host taxonomic assignment. The top and bottom of each box represent the 25th and 75th percentiles, and the center line indicates the median. The upper and lower whiskers of each box represent the maximum and minimum values detected in the three biological replicates, respectively. The differences between the pre-and post-desiccation treatments were assessed using a two-sided t-test. The significant difference ($p < 0.05$) is highlighted by an asterisk (*). The exact p-value is 0.03. **b**, **c** The correlations between VPH and host abundance in soils pre- and post-desiccation, respectively. The regions shaded in dark gray along the lines represent the 95% confidence level intervals in linear regression. The significance of the regression analysis was determined by an F-test with the adjusted $R^2$ and p-value to estimate the strength of the relationship. Source data are provided as a Source Data file.

hosts of soil viruses have been challenging due to the complexity of the soil environment, coupled with the vast diversity of largely uncharacterized soil viruses[2,22]. As a result, the current state-of-the-science has been to bioinformatically predict soil viral hosts from metagenomes[5–7,16]. For example, CRISPR spacer matching is one of the most widely used bioinformatic methods for phage host prediction in the absence of direct evidence[15,16]. This approach relies on the storage of immunity memories to prior viral infections in spacer arrays in the host DNA[17]. CRISPR spacers can be conserved for years in prokaryotic genomes as exact matches to the protospacer targets from past infections[23]. As a result, the number of spacers that the prokaryotic hosts acquire and maintain can range from 10 to 100 and the majority of the spacers acquired do not represent current viral infections[24]. This may partly explain why there were no overlaps in phage-host links predicted by the CRISPR spacer matching method and those detected by Hi–C sequencing. Therefore, although CRISPR spacer matching and other current bioinformatic approaches are valuable for revealing potential phage–host interactions[5–7,16], they are deficient in the identification of specific hosts that phage are infecting at the time of sampling. The Hi–C approach provides an important advance for soil microbial ecology by chemically linking phages to their bacterial host cells and generating experimental evidence of infection at the time of sampling. We acknowledge, however, that both of these approaches are subject to under-sampling considering the high diversity of soil phages[2]. With deeper sequencing, we anticipate that there would be some overlap between phages detected using the two approaches. Currently, the combination of both methods is complementary and provides better coverage of soil phages than using only one approach.

We also acknowledge that the Hi–C approach does not detect all possible phage-host pairs in complex soil samples because it relies on the extraction of host cells and their associated phage from the bulk soil matrix prior to sequencing. This could be a reason for the lower relative abundance of Hi-C-detected host-associated phages in this soil study (5.3% to 15.0%) compared to that in other systems that have used this approach to date; such as the human gut (~94%, assigned to MAG)[14] and wastewater treatment plants (~37%)[25]. The lower proportion of infected hosts in soil implies that a large proportion of the bacterial hosts were not extracted from the soil matrix. Alternatively, a higher fraction of the soil phage community consists of free phages that have not infected their hosts. Regardless of these shortcomings, the findings reported here emphasize the value of the Hi-C approach for identifying real-time phage-host pairs in soil. These findings also revealed the impacts of changes in soil conditions on phage infections of the soil microbiome.

Using the Hi-C sequencing approach, we provide direct evidence of the response of host-associated phages to soil drying. As soil dries and aqueous habitats become more sparse and fragmented[26], free soil phages can irreversibly bind to the dried soil particles and thus become inactive[27,28]. Therefore, we hypothesized that in desiccated soil, phages residing inside their hosts would be preferentially retained in a lysogenic lifecycle. In support of this hypothesis, the host-associated phages in post-desiccation soil had lower levels of transcriptional activity, suggestive of lysogeny. In addition, the higher average VPH in the soil after drying indicated that a higher proportion of hosts were infected by phage, and these were presumably lysogenic. By contrast, in pre-desiccation soil the phage infections resulted in a

significant decrease in host abundances, suggesting that the hosts were lysed by phage via the lytic cycle. Together these results suggest that as soil dries, there is a transition of phage lifestyles from primarily lytic to lysogenic. These results support prior studies that reported more lysogenic markers (i.e., genes encoding integrases and excisionases)[5] but fewer viral transcripts in dry compared to wet soil[4]. In this study, the Hi-C approach provides direct evidence of the impact of soil drying on soil phage lifestyles and specific phage host interactions.

Hi-C sequencing revealed shifts in phage-host interactions following a two-week experimental desiccation period. Few overlaps in phage-host pairs were detected pre- and post-desiccation. Both physical and biological factors could contribute to the differences in host selection, and in turn the types of phages that persist in soil under changing moisture conditions. Because soil desiccation leads to a fragmented soil habitat with micro-niches that are not connected by water[26,29], this physical separation limits microbial dispersal and phage-host interactions. This may explain the reduced richness of bacterial host MAGs in post-desiccation soil. However, phage speciation often overcomes barriers to reproduction when phages and hosts are spatially separated[30,31]. Therefore, soil desiccation may promote niche differentiation and phage diversification as supported by our finding of a higher richness of the host-associated phages after soil drying. We note that our experimental design and data cannot address the possibility that changes occurred over time independent of the moisture treatment, or that desiccation affected the susceptibility of hosts due to changes in outer membrane composition. Other factors that may contribute to the differences in phage-host pairs detected in soils after desiccation include the survival, metabolic activity, and reproductive capacity of different potential host populations. Host metabolism and replication are critical for phage replication. Some phages even encode sigma factors to prevent bacterial dormancy to enhance their ability to replicate with their hosts[32]. Previous studies have reported the responses of different bacterial taxa to changes in soil moisture[33–35]. For example, some members of the *Actinobacteria* are known for their tolerance to low soil moisture and have been shown to be relatively more abundant in drier soils[33–35]. Interestingly, in post-desiccation soils Actinobacteria was one of the dominant phyla with primarily lysogenic infections. These findings support the Piggyback-the-Winner hypothesis[36], suggesting that lysogenic phages favor hosts with fitness advantages (e.g., bacteria that are able to survive under dry conditions and retain higher microbial abundances). Understanding how the changing environment influences phage-host interactions is therefore important for predicting the consequences of soil phages on microbial compositions and functions.

Phage infections influence the soil bacterial community structure. Previous studies have linked high centrality in a community co-occurrence network with biological importance[19]. MAGs that occupied central nodes in the bacterial co-occurrence networks represent the taxa that were presumably central to the soil bacterial community structure (i.e., central taxa). Our finding of phage infections within the central taxa suggests that changes in phage abundances and their lifecycles as a function of soil moisture shifts have impacts on inter-species interactions among soil bacterial communities, even if only a small number of taxa are infected[37–39]. Phage predation can influence nutrient availability and affect the soil bacterial community structure as well. Host lysis due to viral infection releases cellular material that can be assimilated by other microbes (the process is known as viral shunt[40]). Additionally, phage infection can remodel the host metabolic potential, for example, via lysogenic conversion[41,42] and expression of auxiliary metabolic functions[43]. Therefore, phage predation of the central taxa may have outsized effects on the whole soil bacterial community[44].

The detection of phages infecting multiple hosts (viral generalists) using Hi-C sequencing is intriguing. Although putative viral generalists have been reported in various ecosystems[5,16,45], they were primarily identified based on CRISPR spacers and genomic similarities[5–7,16], except for few studies that experimentally isolated phage generalists with host ranges across bacterial phyla such as from Lake Michigan[46]. Applying Hi-C sequencing provided us with an opportunity to use direct experimental evidence to assign viral generalists in soil. We acknowledge that there could be limitations in coverage that mask the true extent of soil phage-host interactions when using the Hi-C approach. As the cut-offs we used for clustering phages were at the species rank[47], the vOTUs that were associated with multiple de-replicated host MAGs clustered phages that were phage generalists. The detection of 15 unique viral contigs linked to multiple dereplicated MAGs provides direct evidence of the presence of phage generalists in soil. The emergence and persistence of phage generalists are suggested to be a result of phage adaptation to diverse bacterial communities. A previous study showed that phage generalists evolved as early as four days after the ancestor phages were co-cultured with the original *Escherichia coli* hosts and the non-permissive strains[48]. Therefore, the detection of phage generalists in our study suggests phages actively evolve and adapt to the bacterial communities in soil. Furthermore, we found that phage generalists had a higher richness after soil drying and were relatively more abundant than phage specialists (phages infecting one host). These results suggest that having a broad host range provides a competitive advantage to phages to enable them to exploit diverse hosts and maximize their fitness in desiccated soil[12]. Similarly, phage generalists were detected in some of the most abundant vOTUs of Arctic peat soil under stressful anoxic conditions[45]. Our study provides the direct evidence of the presence of phage generalists in soil using metagenomes and points to the opportunities for more quantitative measures of the fitness of soil phage generalists and specialists as well as fitness trade-offs of being a generalist are needed in future studies.

This is the primary application of Hi-C sequencing to a soil environment to capture ongoing viral infections and to determine specific viral host interactions. This study also provides direct experimental evidence of the presence of viral generalists that, to this date, had mainly been predicted bioinformatically. By applying Hi-C together with other DNA and RNA sequencing approaches, this study provides insights into the impact of soil drying on phage-host interactions and the downstream impacts on bacterial community interactions. Although this study reveals the promise of the Hi-C approach for detecting phage-host interactions in complex samples, such as soil, it has the potential to be further improved with additional experimental optimization. This will allow future studies to further explore the application of the Hi-C approach for detecting viral infections in other soil systems and natural environments. We note that the Hi-C approach is also applicable for the detection of environmental viruses that are potential biothreats. This study demonstrates that Hi-C sequencing is applicable to soil and the results have value for predicting the impact of climate change on soil viruses and the ensuing ecological consequences.

## Methods

### Soil sampling and incubation

Soils were collected in collaboration and compliance with the Washington State University Irrigated Agriculture Research and Extension Center. Surface soils (0-20 cm deep, 5 cm diameter cores) were collected from 16 randomly selected locations immediately adjacent to the Tall Wheatgrass Irrigation Field Trial experiment in Prosser, WA (46°15′04″N, 119°43′43″W) in June 2020. The average summer temperature is 30 °C and the average annual precipitation is 180 mm[5]. Samples were transported to the Pacific Northwest National Laboratory (PNNL) on ice and stored at 4°C until processing. Soils were homogenized on ethanol-sterilized 4 mm sieves, and rocks were removed. Gravimetric soil moisture content was determined by drying

three subsamples of 10 g of soil in an incubator at 60 °C for 24 h. Water holding capacity was determined by saturating 5 g of soil, weighing after 2 h when soils were completely drained, and drying at 60 °C for 24 h. The weight difference between saturated and desiccated soils was used to calculate the water holding capacity of the soil.

Sixty grams (dry weight soil equivalent) were weighed into autoclaved 4 oz jars, and the soils were brought to 75% water holding capacity by addition of sterile deionized water. Three replicate incubations were set up for each of the two sampling time points (*n* = 6). First, the soils were preincubated for one week during which time they were gradually exposed to increasing temperature until the target incubation temperature of 30°C average summer temperature (https://www.usclimatedata.com/climate/prosser/washington/united-states/uswa0355) was reached. The incubation temperatures were: 10 °C on day 1, 15°C on day 3, 25°C on day 5 and 30°C on day 7. During the pre-incubation water was added to the soils daily to maintain them at 75% water holding capacity by mass.

Once at 30°C, three samples were harvested representing the pre-desiccation soils. The remaining three incubation jars continued to incubate at 30°C, without any further moisture additions. Moisture loss was monitored by mass, until complete desiccation was achieved after 14 days. Then the final three replicate samples were harvested, representing the post-desiccation soils. The incubation of soil samples under field-relevant conditions (i.e., 30°C for 2 weeks) mimicked the natural soil drying process in hot weather typical of Prosser, Washington summers. The six soil samples, with three from each timepoint, were stored at -80°C until being processed for DNA and RNA extractions.

### DNA and RNA extraction, shotgun library generation and sequencing

Frozen soil samples (*n* = 6, three for each time point) were handled on dry ice during aliquoting/transfer to extraction tubes. DNA was extracted from 0.25 g of individual soil samples using the ZYMO-biomics DNA miniprep kit (Zymo Research, Irvine, CA, USA), and RNA was extracted from 2 g of individual soil samples using the PowerSoil RNA isolation kit (Qiagen, CA, USA), each in accordance with the manufacturers' instructions. The genomic DNA extracted from the six samples was quality-checked using Qubit BR (Invitrogen, Waltham, MA, USA) and shipped to Phase Genomics on dry ice. The paired-end deep sequencing libraries were prepared using ProxiMeta library preparation reagents (Phase Genomics, Seattle, WA). Sequencing was performed on an Illumina NovaSeq generating an average of 177 million PE150 read pairs.

The extracted RNA was treated with Turbo DNase (Invitrogen, Waltham, MA, USA) followed by clean up with a Zymo RNA Clean and Concentrator Kit purification kit (Zymo Research, Irvine, CA, USA). The resulting RNA was quality checked using an Agilent RNA 6000 Nano kit (Agilent, Santa Clara, CA, USA) and quantified using a Qubit (Invitrogen, Waltham, MA, USA). The RNA extracted from the six samples was sent to the Joint Genome Institute (JGI) to be sequenced using the standard metatranscriptome workflow (https://jgi.doe.gov/user-programs/pmo-overview/project-materials-submission-overview/rna-submission-guidelines/). The metatranscriptome sequencing was performed on an Illumina NovaSeq S4. The sequencing statistics for the shotgun/bulk metagenomes and metatranscriptomes are shown in Supplementary Data 1a and e, respectively.

### Hi−C sequencing

For each replicate, 5 g of soil per time-point were sent on dry ice to Phase Genomics to be processed per their low-biomass protocol (ProxiMetaTM Hi-C Kit Protocol v4.0). In brief, samples were mixed in 25 ml of water and vortexed for 5 min. The tubes were centrifuged at 1000 x *g* for 10 min. to allow sediment to settle. The supernatant was transferred to a new tube and formaldehyde was added to a final

concentration of 1% (v/v). The tubes were incubated at room temperature for 20 min. with occasional gentle mixing by inversion or rotation. Glycine (ProxiMetaTM Hi-C Kit, Phase Genomics, Seattle, WA) was added to a final concentration of 1% (v/v) to quench the cross-linking reaction and the samples were incubated at room temperature for 20 min. with occasional gentle mixing by rotation.

A Hi-C library was created using a ProxiMeta Hi-C Microbiome v4.0 Kit (Phase Genomics, Seattle, WA) which is the commercially available version of the Hi-C protocol[49]. Following the manufacturer's instructions, the cross-linked DNA extracted from each replicate was digested using Sau3AI and MlucI restriction enzymes (ProxiMetaTM Hi-C Kit, Phase Genomics, Seattle, WA), and proximity-ligated with biotinylated nucleotides (ProxiMetaTM Hi-C Kit, Phase Genomics, Seattle, WA) to create chimeric molecules composed of fragments from different regions of genomes that were physically proximal in vivo. Chimeric molecules were pulled down with streptavidin beads and processed using the ProxiMeta library preparation reagents (Phase Genomics, Seattle, WA). The Hi-C metagenomes were sequenced on an Illumina NovaSeq. The quality matrices demonstrated strong proximity signals in each Hi-C library were included in Supplementary Data 1C.

### Shotgun sequence processing and viral contig detection

Shotgun sequencing reads were trimmed, quality-filtered and normalized using fastp (v0.20.1)[50]. The resulting sequence data from replicate samples were co-assembled using MEGAHIT (v1.2.9) using the meta-large preset option. Contigs longer than 1 kb were retained for downstream analysis (Supplementary Data 1b).

Metagenome-assembled contigs with lengths longer than 5 kb were screened for viral sequences (Supplementary Data 1b). To reduce the possibility of introducing false positives, multiple viral bioinformatic detection algorithms were used, incorporating stringent or suggested cut-offs. These tools included VirSorter (v2, a minimum score of 0.5)[51], VIBRANT (v1.2.1, with 'virus' tag predicted by neural network model)[52], DeepVirfinder (v 2020.11.21, score > 0.9 and *p* < 0.05)[53], and CheckV (v0.7.0) for assessing viral genome quality[54]. Contigs were classified as viral if they met the following criteria: (1) classified as viral by at least two of the three viral detection tools, or (2) classified by one viral detection tool as complete or with high-to-medium genome quality. Applying this voting strategy to the results from multiple tools is intended to avoid potential limitations in one tool and minimize the detection of false positives.

### Hi-C sequence processing

Hi-C reads were quality-controlled via the same method that was applied to the shotgun metagenome. The quality-filtered Hi-C reads were aligned to the de-novo assemblies obtained from the paired shotgun metagenomes following the Hi-C kit manufacturer's recommendations (Phase Genomics, Seattle, WA, https://phasegenomics.github.io/2019/09/19/hic-alignment-and-qc.html). Briefly, reads were aligned using BWA-MEM (v0.7.17)[55] with the -5SP options specified to reduce the secondary and alternative mappings. SAMBLASTER (v0.1.24)[56] was used to flag PCR duplicates, which were later excluded from the analysis. Alignments were then filtered with samtools (v1.9)[57] using the -F 2304 filtering flag to remove non-primary and secondary alignments.

### Metagenome binning and clustering

Metagenome deconvolution was performed with ProxiMeta[58,59], which implements a graph-based clustering algorithm using the shotgun metagenome assembly and Hi-C mapping as input to generate metagenome-assembled genomes (MAGs). Briefly, Hi-C mappings were filtered with a minimum score of 20 after removing the Hi-C metagenomic paired-end reads that were non-uniquely mapped to contigs or mapped to the same contig (length of the contig > 1 kb).

The filtered Hi-C mappings were then used to generate a contig-contig interaction network. These contigs were further clustered into genome bins using a proprietary Markov chain Monte Carlo (MCMC)-based algorithm[60]. The genome bins or MAGs were assessed for quality using CheckM (v1.2.0)[60]. To visualize the grouping of the linked contigs, the Hi-C contact maps were generated using the quality-filtered assemblies and the reads mapping files (BAM files) via bin3C (v0.11)[61,62] and shown in Figure S2.

MAGs obtained from each replicate after the deconvolution process were quality-filtered (< 9% contamination) and dereplicated at 99% identity using dRep (v3.4.0)[63]. Taxonomies of the unique MAGs were assigned by CheckM (v1.2.0, lineage-wf) and validated using the Genome Taxonomy Database Toolkit (GTDB-Tk, v2.1.0, classify-wf). The abundance of the MAG was estimated by average read depth normalized by length.

### DNA viral sequence clustering, and taxonomy assignment

To compare the richness of viral communities, the quality-filtered viral contigs were clustered into species-level equivalent vOTUs at 95% of average nucleotide identity (ANI) and 85% of alignment fraction (AF) as recommended previously[47]. A greedy centroid algorithm was applied in clustering as described in the published workflow (https://github.com/snayfach/MGV/blob/master/ani_cluster/README.md)[64]. In brief, viral contigs were first sorted by length and the one with the longest length was selected as the centroid or representative sequence of a new cluster. The rest of the viral contigs were scanned and assigned to an existing cluster if they met the cutoffs of 95% ANI and 85% AF. A proteomic tree of the vOTU representative sequences was constructed using a neighbor joining method via ViPTreeGen (v1.1.2, a command-line version of ViPTree)[65]. In brief, the pairwise similarities of the vOTU representative sequences were computed based on the tBLASTx results. The distance between the viral sequences was demonstrated in a dendrogram that was further clustered using the neighbor-joining algorithm.

For taxonomic assignments of the detected viral contigs, the representative sequences of the vOTUs were clustered with genomes deposited in the INfrastructure for a PHAge REference Database (INPHARED) using the same clustering algorithms and cutoffs as mentioned above (https://github.com/snayfach/MGV/blob/master/ani_cluster/README.md)[64], and with NCBI Refseq viral genomes (v201) using vConTACT2 (v0.9.19, default settings). Due to the high sequence diversity and novelty of the detected viral contigs from soils, none of the representative sequences were able to be clustered with reference viral genomes using both ANI and protein sharing matrix (implemented in vConTACT2) methods. Additionally, we exhausted another approach to assigning viral order and family using amino acid homology searches against viral reference sequences in TrEMBL[66] that is implemented in the Demovir workflow[67]. Nearly half of the viral contigs remained unclassified with the rest assigned to Family levels: *Myoviridae*, *Podoviridae* and *Siphoviridae*. According to the latest release of the International Committee on Taxonomy of Viruses (ICTV) (Release 2021), these three families were recommended to be abolished and assigned to class Caudoviricetes[68,69]. Thus, these viral contigs were annotated as Caudoviricetes.

### Calculation of DNA viral coverage in metagenomes and metatranscriptomes

Per-sample coverage depth and breadth were calculated for each contig using BBMap (v 38.34)[70]. Contigs with more than 50% of their genome mapped (i.e., breadth of the coverage) were considered to be positively detected in the respective metagenome[5,71]. The richness of the viral community was represented by the number of vOTUs that were positively detected. The abundances of the contigs with more than 50% breadth of coverage were estimated using the average base coverage mapped by the reads of the respective metagenome (i.e.,

depth of the coverage). The abundance estimates of the contigs with a breadth of coverage lower than 50% were recorded as zero. The abundance estimates were then normalized by the number of assembled reads for each metagenome.

The quality-filtered metatranscriptome reads were mapped to the viral contigs that were positively detected in metagenomes (described above) from the same sample by BamM (v1.7.3, bamm make). The read mapping was filtered at 95% of identity and 80% of alignment fraction (BamM v1.7.3, bamm filter). Transcription levels were estimated by the average base coverage of the viral sequences using samtools (v1.9, samtools depth)[72] normalized by the total counts of metatranscriptome reads per sample. The richness of the viral community that was potentially transcriptionally active was represented by the number of vOTUs that were detected by the metatranscriptome.

### Phage−host network construction and analysis using Hi−C data

The average viral copy per host cell (VPH) was estimated using Hi−C link count ($L(v)$ representing Hi−C links of the phage with all possible hosts, $L$ representing Hi−C links of each phage-host pair), host-associated vOTU abundance ($V$), and host abundance ($H$) (Eq. 1)[73]. The phage-host pairs linked by Hi−C reads were screened by two rounds of filtering to remove false positives.

$$VPH = \frac{V}{H}\frac{L}{\sum L(v)} \qquad (1)$$

The first-round filtering criteria[73] include the following: (1) at least two Hi−C reads linking the phage and host MAG, (2) a phage-host connectivity ratio ($R'$) of 0.1, and (3) intra-MAG connectivity of 10 links. The phage-host connectivity ratio ($R'$) was calculated using a Hi−C connectivity density of the phage-host pair ($D_{VH}$) and of the MAG to itself ($D_H$) after normalized by VPH (Eq. 2). The Hi−C link count per kb$^2$ of sequence was used to estimate the connectivity density.

$$R' = \frac{D_{VH}}{D_H}\frac{H\sum L(v)}{VL} \qquad (2)$$

The second round of filtering was based on the threshold value from a receiver operating characteristic (ROC) curve. The optimal threshold is the value which maximizes the number of phages with at least one host, while also removing the largest number of phage-host links. Additional filtering removed the phage-host linkages with average counts that were less than 80% of the highest count of the linkages of the same viral contig. Finally, some phages which interact with a high fraction of host MAGs in the sample were further evaluated and adjusted to control for phage sequences which are rich in false positive interactions.

The phage-host infection pairs identified from Hi-C sequencing were further grouped by combing the viral contigs that were clustered in the same vOTU with host bins that were more than 99% identical (indicated by dRep, v3.4.0)[63]. As a result, we generated unique phage-host infection pairs and assessed whether those pairs were consistently detected across replicates under each treatment (Supplementary Data 1D). The vOTUs that were assigned to more than one unique host bin and those assigned to only one unique host bin by Hi-C sequencing were categorized as vOTUs with multiple hosts and vOTUs with one host, respectively. The rest of the vOTUs that were not captured by Hi-C sequencing or not host-associated were classified as no host detected.

### Phage-host links predicted by the CRISPR spacer matching method

The phage-host links were also independently predicted using the CRISPR spacer matching method. The CRISPR spacers of each MAG were retrieved using CRISPRCasFinder (v3.1.0) with the options of

-gscf and -cas. The viral contigs screened from shotgun metagenome-assembled contigs were formatted into a BLAST database. All the CRISPR spacers were queried against the viral contig database using the task option of blastn-short. The additional parameters including a percent identity of 95%, one maximum target sequence and at most one mismatch allowed were applied to filter the matching result[16].

## Microbial co-occurrence network

Because we had metatranscriptomic data from a wider range and number of samples than metagenomic data, the microbial co-occurrence networks were constructed from metatranscriptomic data to maximize the number of input samples. The *argS* gene was selected as the proxy because it is a housekeeping gene that is single-copy in most organisms and was present in a majority of the MAGs. The depth of coverage for transcripts of the *argS* gene was converted to relative abundance, and correlation networks were inferred using Pearson's correlation coefficient, Context Likelihood of Relatedness (CLR)[20] or GENIE3[21]. Three methods were chosen for a more comprehensive analysis of bacterial community structure of the studied soil. Each of these chosen methods had elements that spoke to its inclusion. Pearson is a common network inference method here that has been used previously for microbial co-abundance networks[74]. CLR includes an element of context, using the resulting network matrix to better report putative edges[20] and GENIE3 was found to be highly accurate in network inference[75,76]. For CLR networks, the top 200 edges were collected (representing the edges with values in the 95th percentile). For GENIE3 a similar edge cutoff was used, but since GENIE3 is a directional network method this led to slightly more edges (311). For Pearson correlation coefficient networks, any edge with an absolute value of > 0.8 was used (186 edges). Once each network was inferred, Cystoscape[77] was used to determine betweenness and degree centrality values for each node. Nodes (bins) with high centrality were then compared to those bins that were targeted by phage.

## Statistical analysis

T-tests were used to assess the significance of the differences in viral richness, relative abundances, transcription levels and potential life strategies (lytic or lysogenic) of viral communities in the soil replicate samples before and after the soil drying treatment. A linear regression model with a 0.95 confidence level was applied to test the correlations between the average viral copies per host cell and the estimated abundances of hosts. The statistical tests were performed via RStudio (v2022.07.1) with R (v4.2.0). Differences between each comparison or correlation with a p-value less than 0.05 were considered as significant. The experiment was run using three replicate samples for each treatment for generating reproducible results.

## Reporting summary

Further information on research design is available in the Nature Portfolio Reporting Summary linked to this article.

## Data availability

The raw or large processed data generated in this study have been packaged and publicly available at DataHub (https://data.pnnl.gov/group/nodes/dataset/33511) as well as NCBI under BioProject PRJNA1006511. Four data packages include: (1) raw sequences of the six shotgun metagenomes (https://doi.org/10.25584/1922087, https://data.pnnl.gov/group/nodes/dataset/33337, NCBI: SRR25682926-SRR25682930), (2) all the shotgun metagenome-assembled contigs (>1 kb), MAGs and the dereplicated viral contigs (https://doi.org/10.25584/1922088, https://data.pnnl.gov/group/nodes/dataset/33338, NCBI: JAWMQX000000000 and JAWMQY000000000 for the contigs assembled from soils with pre- or post-desiccation treatment), (3) the unique phage-host pairs detected by Hi-C (https://doi.org/10.25584/1922090, https://data.pnnl.gov/group/nodes/dataset/33339), (4) the

quality-filtered metatranscriptomes (https://doi.org/10.25584/1922091, https://data.pnnl.gov/group/nodes/dataset/33340, NCBI: SRR25916027- SRR25916064), and (5) raw sequences of the six Hi-C metagenomes (https://doi.org/10.25584/1970740, https://data.pnnl.gov/group/nodes/dataset/33511, NCBI: SRR25689209-SRR25689214). A detailed description of each data package is in Supplementary Data 1. Source data are provided with this paper.

## Code availability

The R codes used to plot the main figures are available on GitHub (https://github.com/Ruonan0101/SFA_Hi_C_MS)[78] with no restriction to access.

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

## Acknowledgements

This program is supported by the U. S. Department of Energy, Office of Science, through the Genomic Science Program, Office of Biological and Environmental Research, under FWP 70880 (K.S.H). PNNL is a multi-program national laboratory operated by Battelle for the DOE under Contract DE-AC05-76RLO 1830. A portion of this research was performed on a project award (project number: 50978, J.E.M) under the FICUS program and used resources at the DOE Joint Genome Institute and the Environmental Molecular Sciences Laboratory, which are DOE Office of Science User Facilities. Both facilities are sponsored by the Biological and Environmental Research program and operated under Contract Nos. DE-AC02-05CH11231 (JGI) and DE-AC05-76RL01830 (EMSL). We thank PHASE GENOMICS for their technical support for the Hi-C sequencing and Sheryl L. Bell for facilitating the experiment and providing technical details of RNA sample preparation.

## Author contributions

J.K.J., K.S.H., M.R.D., and R.W. designed the Hi–C experiment. M.L.S. collected the soil samples from the field. M.L.S. and M.S.L. set up the soil incubation experiment and collected samples for DNA and RNA extraction. M.R.D. processed soil samples for DNA extraction and coordinated sample exchange with PHASE GENOMICS. PHASE GENOMICS performed the Hi–C sequence analysis. R.W. performed the bioinformatics analyses, including detection of viruses, MAG dereplication and annotation. W.C.N. assembled and analyzed the metagenomes. R.S.M. constructed the microbial co-occurrence networks. K.S.H., J.K.J., and J.E.M. obtained funding for the study. All authors contributed to writing the manuscript.

## Competing interests

The authors declare no competing interests.
