## [Peer Review File · Nature Communications]

REVIEWER COMMENTS

Reviewer #1 (Remarks to the Author):

The publication by Wu et al., investigate microbe-virus interactions in soil and the impact of moisture on those interactions. For this purpose, they take soil samples and study them in laboratory under dry and wet conditions. They then used a combination of metagenomic, metatranscriptomic and proximity ligation (3C/HiC) methods to study those samples.

They found an increase of viral copies in dry conditions associated with a lower transcriptional activity compared to wet conditions. Their results suggest that lytic cycles are favored under wet conditions while lysogenic are prevalent in dry conditions. The findings could be interesting, and the experimental design is original.

However, the overall conclusion of the authors appears difficult to be generalized as it only concerns 19 of the 479 vOTUs detected (between 5.3% to 15% of the total viral sequence abundance detected in the samples). In general, the results appear weak, and I would suggest lowering the findings and to discuss this in the conclusion. Some points also need to be clarified and some important controls are lacking.

Major Comments:

- First of all, I could not find a link to have access to raw proximity ligation data. It is mandatory to provide an access to these data to assess their quality. The link provided (<https://data.pnl.gov/group/7/nodes/publication/34802>) is not valid.

- In regards of reproducibility and open science, authors should also provide their assemblies by submitting them to NCBI or other databases. By doing this, the scientific community will be able to reproduce the results.

- Line 113: "We identified 148 unique MAGs (< 9% contamination), spanning nine bacterial phyla". What proportion of the assemblies these 148 MAGs represent? it is important to know this statistic in order to appreciate the generalization of the results.

- Very few statistics are provided concerning the data (size of the assemblies, mapping back rate, proximity ligation). It appears extremely difficult to evaluate the quality of their HiC data (raw reads, mapping rate, 3D ratio, noise signal). Indeed, no matrices are provided, and it is impossible to evaluate

the noise signal encompass in their HiC data. Authors should, at least, provide a matrix of the different MAGs obtained (<https://journals.asm.org/doi/10.1128/MRA.01523-19>).

- Authors should also compare Shotgun and HiC data. Is there particular biases in term of contigs coverage concerning the different libraries? a PCA analysis or other types of analysis in order to compare the different libraries would allow to reinforce the reproducibility of their results. How the different vOTUs behave in general depending on the different conditions? how the replicate overlap?

- Line 98 : “Hi-C sequencing resulted in 118 unique, high-confidence phage-host pairs”. From how many reads authors have obtained this number of linkages? This number appears very low.

- In the same way, authors state in the Methods section: “The virus-host pairs linked by Hi-C reads were subjected to a series of aggressive filtering to remove false positives. The first-round filtering criteria include the following: 1) at least two Hi-C reads linking the virus and host MAG ...” . How a filter at only 2 contacts can be an aggressive filtering step ?

- Line 99: “Viruses belonging to 19 of the 479 detected vOTUs were assigned to their respective host metagenome assembled genomes”. If I understand well the data and the table (that are not easy to understand); this 19 vOTUs are, for most of them, assigned as low-quality draft by VIBRANT. Therefore, it appears difficult to propose general rules of phages behavior solely based on 19 contigs representing low-quality phages genomes.

- Is there a possibility that the characterized bacterial host are the most abundant organisms of their samples? is there a correlation between the number of host and the host and/or the virus abundancy? indeed, highly abundant sequences will be more prone to noise signal and, therefore, to false positive detection.

- Line 168: “We calculated the average viral copies per host (VPH) to represent the degree of phage infection per host population.” While their calculation based on HiC links could be interesting, it is somehow peculiar that this value varies from 0 to 1.5 ... I would have expected a much higher value for lytic phages. Could the authors perform the same kind of calculations using Shotgun reads and compare those values?

Minor comments:

- Line 95: ref 23 is not the appropriate one to cite HiC applied to metagenomic as it is the first HiC publication on Human

Martial MARBOUTY

Reviewer #2 (Remarks to the Author):

The study showcases how Hi-C sequencing data can be used to explore virus-host linkages in soils. According to the text, the aim of the experiment was to explore phage lifestyle comparing wet vs dry soils. However, the experimental design included two sampling points: before and after a 14-days incubation period. The replicates (n=3) sampled at the end were not watered (dry soil treatment) and compared with the replicates (n=3) sampled at the beginning when water holding capacity was 75%. Ideally, a control condition made of wet soil should be considered and sampled at the end of the experiment together with the dry soil treatment. Without that, the experiment was not accounting for the incubation time factor. Therefore, the conclusions presented to the readers may not be fully supported by the results observed.

To my knowledge this is the first study to apply Hi-C sequencing to soil. In terms of sequencing methods, may be worth exploring the data obtained to highlight methodological aspects and possibilities of using Hi-C sequencing for soil communities. So it may be possible to restructure the focus of the message, to tone down the conclusions and avoid direct dry vs wet comparisons given the experimental limitation, and instead try to emphasise the methodological aspect of analysing Hi-C sequencing applied to soil environmental DNA. Below I provide comments and suggestions with the intention to help to improve the study.

Would be helpful to improve Fig S1 to make more clear the experimental and computational steps used, including how each dataset (metagenome, metatranscriptome and Hi-C) was used in the study. Maybe this figure could be moved to main text.

It is mentioned about computational methods currently used to predict virus-host linkages (L51-54). Why not applying those to the data to help the community providing evidence on how much virus-host

assignments can be improved with Hi-C approach? Let's say for example that by running these computational predictions for the virus-host linkages detected with Hi-C you get no host assigned to most (or few) of these viruses(?). Including this kind of analysis will help to emphasise the point mentioned in the Introduction, and exploring ideas like that may help to find new focus for the message.

The metagenomes were obtained from 0.25g of soil while the Hi-C sequencing was obtained from suspensions of 5g of soil using water. So discrepancies are expected when comparing diversity of viruses and host between metagenomes and Hi-C sequencing. They can arise mainly i) because of the difference in the amount of soil sampled (0.25g vs 5g), and ii) because only extractable viruses and host could be captured by Hi-C sequencing. Would be important to discuss this limitation.

How the viral contigs from Hi-C sequencing data were identified? Was it in the same way as the contigs from the metagenomes? MAGs from metagenomes were not recovered?

How was lysogeny inferred?

The manuscript might benefit of using the virocell concept.

L24: (and elsewhere): Is it 100% safe to generalise using the term 'infection' based only on Hi-C? Would be important to discuss that.

L367: Provide more details about the method used to recover MAGs.

L379: Do you mean 'min' score of 0.5?

L429-430: Wouldn't it be easier for the reader if V and H were simply called as vOTU and host abundances, respectively?

L462: clarify here that abundance refers to MAG activity/transcript?

L388: please clarify the tools used for the clustering step (was it the same as in L396?)

L391: provide more details on how the vOTU proteomic tree was constructed.

Figure 4: consider converting back to natural numbers as they are more biologically informative than log values. Have you also tried to correlate the activity/transcripts?

**REVIEWER COMMENTS**

Reviewer #1 (Remarks to the Author):

The publication by Wu et al., investigate microbe-virus interactions in soil and the impact of
moisture on those interactions. For this purpose, they take soil samples and study them in
laboratory under dry and wet conditions. They then used a combination of metagenomic,
metatranscriptomic and proximity ligation (3C/HiC) methods to study those samples.
They found an increase of viral copies in dry conditions associated with a lower transcriptional
activity compared to wet conditions. Their results suggest that lytic cycles are favored under wet
conditions while lysogenic are prevalent in dry conditions. The findings could be interesting, and
the experimental design is original.

**Reply:** We thank the reviewer for this encouraging comment and all the valuable suggestions
below.

1. However, the overall conclusion of the authors appears difficult to be generalized as it only
concerns 19 of the 479 vOTUs detected (between 5.3% to 15% of the total viral sequence
abundance detected in the samples). In general, the results appear weak, and I would suggest
lowering the findings and to discuss this in the conclusion. Some points also need to be clarified
and some important controls are lacking.

**Reply:** We added a section to the discussion to address this question as follows:

“Therefore, although CRISPR matching and other current bioinformatic approaches are valuable
for revealing potential phage-host interactions^{5-7,16}, they are deficient in the identification of
specific hosts that phage are infecting at the time of sampling. The Hi-C approach provides an
important advance for soil microbial ecology by chemically linking phages to their bacterial host
cells and generating experimental evidence of infection at the time of sampling. We
acknowledge, however, that both of these approaches are subject to under-sampling considering
the high diversity of soil phages². With deeper sequencing, we anticipate that there would be
some overlap between phages detected using the two approaches. Currently, the combination of
both methods is complementary and provides better coverage of soil phages than using only one
approach. We also acknowledge that the Hi-C approach does not detect all possible phage-host
pairs in complex soil samples because it relies on the extraction of host cells and their associated
phage from the bulk soil matrix prior to sequencing. This could be a reason for the lower relative
abundance of Hi-C-detected host-associated phages in this soil study (5.3% to 15.0%) compared
to that in other systems that have used this approach to date; such as the human gut (~94%,
assigned to MAG)¹⁴ and wastewater treatment plants (~37%)²⁵. The lower proportion of infected
hosts in soil implies that a large proportion of the bacterial hosts were not extracted from the soil
matrix. Alternatively, a higher fraction of the soil phage community consists of free phages that
have not infected their hosts. Regardless of these shortcomings, the findings reported here
emphasize the value of the Hi-C approach for identifying ‘real-time’ phage-host pairs in soil for
the first time. These findings also revealed the impacts of changes in soil conditions on phage
infections of the soil microbiome.”

We also added a sentence to the conclusions as follows: “Although this study reveals the promise
of the Hi-C approach for detecting phage-host interactions in complex samples, such as soil, it
has the potential to be further improved with additional experimental optimization.”

We provided additional information on data quality control as requested in Supplementary
Tables 1 and 5 and a new Supplementary Table 4.

Major Comments:

2. - First of all, I could not find a link to have access to raw proximity ligation data. It is
mandatory to provide an access to these data to assess their quality. The link provided
(<https://data.pnl.gov/group/7/nodes/publication/34802>) is not valid.

**Reply:** We updated the link that can be used for external access in the revised manuscript
(<https://data.pnml.gov/group/nodes/publication/33336>). All the datasets are minted with DOIs as
shown in Supplementary Table 1.

3. - In regards of reproducibility and open science, authors should also provide their assemblies
by submitting them to NCBI or other databases. By doing this, the scientific community will be
able to reproduce the results.

**Reply:** Thank you for the comment. We added all the shotgun metagenome assemblies (>1 kb)
to DataHub (a long-term data repository, similar to NCBI submission) and assigned a DOI to the
dataset. The information is updated in the Data Availability section and Supplementary Table 1.

4. - Line 113: “We identified 148 unique MAGs (< 9% contamination), spanning nine bacterial
phyla”. What proportion of the assemblies these 148 MAGs represent? it is important to know
this statistic in order to appreciate the generalization of the results.

**Reply:** To address this question, for each sample we calculated the percentage of detected
contigs (> 1kb, read coverage > 0 in the replicate metagenome) that were binned into MAGs.
The percentages range from 0.443% to 0.651%. Other published metagenome studies of soils
have reported similar levels of success in extracting MAGs (e.g., doi:10.3390/genes10060424,
doi: 10.1128/mra.00804-22, and doi: 10.1038/s43705-022-00100-z). The statistics included in
the table below were added to Supplementary Table 1.

Sample	Total bins	Total binned contigs	% of total detected Contigs
Post-desiccation-SM297bins	57	16841	0.651%
Post-desiccation-SM306bins	35	11463	0.443%
Post-desiccation-SM317bins	56	13410	0.518%
Pre-desiccation-SM324bins	54	12547	0.524%
Pre-desiccation-SM330bins	62	14238	0.594%
Pre-desiccation-SM335bins	74	15276	0.637%

5. - Very few statistics are provided concerning the data (size of the assemblies, mapping back rate, proximity ligation). It appears extremely difficult to evaluate the quality of their HiC data (raw reads, mapping rate, 3D ratio, noise signal). Indeed, no matrices are provided, and it is impossible to evaluate the noise signal encompass in their HiC data. Authors should, at least, provide a matrix of the different MAGs obtained (<https://journals.asm.org/doi/10.1128/MRA.01523-19>).

Reply: Thanks for this suggestion. We added a table containing the relevant statistics of each link. A supplementary table (Supplementary Table 4), in addition to Supplementary Table 5, was added to the revised manuscript. The statistics include the raw and processed information of contig read depths in the Hi-C data, the linked contig read depths, intra- and inter-linkage read counts and density (with and without normalization) detected in all replicate samples as suggested by the reviewer.

For the other statistic details of shotgun metagenomes, de-novo assemblies and Hi-C metagenomes, please refer to Supplementary Table 1. Each dataset is publicly available and assigned with a DOI.

For statistical details of MAG, please refer to Supplementary Table 6.

6. - Authors should also compare Shotgun and HiC data. Is there particular biases in term of contigs coverage concerning the different libraries? a PCA analysis or other types of analysis in order to compare the different libraries would allow to reinforce the reproducibility of their results. How the different vOTUs behave in general depending on the different conditions? how the replicate overlap?

Reply: We appreciate this comment and would like to emphasize that one vOTU or one assembled viral contig represents a unique viral population. In principle, Hi-C sequencing detects the viral populations that contain host-associated viruses. The other reviewer also brought up a nice point that these hosts are water-extractable microbes. Shotgun sequencing, ideally, detects all viral populations (i.e., both free and host-associated). So, shotgun and Hi-C data recover different though related microbial communities by just considering the principles of the two methods. That is why the results of the shotgun and Hi-C sequencing were not used for direct comparison but used for identifying viral sequences (shotgun) and screening the ones that were host-associated (Hi-C). The comparisons made in this paper were constrained to comparisons between shotgun data (e.g., the whole viral community structure in pre- and post-desiccation soils) and between Hi-C data (e.g., host-associated viruses in pre- and post-desiccation soils).

We added this explanation to the results when introducing Figure 1 (experimental design and workflow) to clarify that Hi-C and shotgun metagenomes detected related but different communities. The sentences read as follows: “To enable the targeted detection of host-associated phages, we applied Hi-C metagenomics to the same soil samples. In contrast to shotgun metagenomics, Hi-C metagenomics was able to identify the subpopulation of the total viral community that entered and infected host cells extracted from the soil. Specific phage-host interactions were captured using the Hi-C approach through chemical cross-linking of the phage

and host DNA molecules that were co-localized within the same cell at the time of sampling
(Fig. 1).”

We did a nonmetric multidimensional scaling (NMDS) analysis as requested by the reviewer. In
Fig. S1, panel a, all viral populations (with and without host-associated viruses) detected in
shotgun data were plotted against the viral populations (that were ones with host-associated
viruses detected) in Hi-C data. The treatment effect was much more profound than the method
difference. The same pattern was observed in panel b where the viral populations with host-
associated viruses detected in shotgun and Hi-C data were plotted.

We added the NMDS plot and results to the revised manuscript. The added text reads as follows:
“Similar to the phage communities detected in the shotgun metagenomes, the host-associated
phage communities in the Hi-C metagenomes were significantly impacted by soil drying ($p \leq$
0.005 , Fig. S1).”

7.- Line 98 : “Hi-C sequencing resulted in 118 unique, high-confidence phage-host pairs”. From
how many reads authors have obtained this number of linkages? This number appears very low.

**Reply:** We added the Hi-C metagenome sequencing statistics to Supplementary Table 1 and all
of the other statistics such as the read depths of each linked contig and linkage density in the
updated Supplementary Table 5.

8. - In the same way, authors state in the Methods section: “The virus-host pairs linked by Hi-C

reads were subjected to a series of aggressive filtering to remove false positives. The first-round
filtering criteria include the following: 1) at least two Hi-C reads linking the virus and host
MAG ...” . How a filter at only 2 contacts can be an aggressive filtering step?

**Reply:** We removed the word of “aggressive” and revised the sentence on Line 481-482 to read,
“The phage-host pairs linked by Hi-C reads were screened by two rounds of filtering to remove
false positives.”

9. - Line 99: “Viruses belonging to 19 of the 479 detected vOTUs were assigned to their
respective host metagenome assembled genomes”. If I understand well the data and the table
(that are not easy to understand); this 19 vOTUs are, for most of them, assigned as low-quality
draft by VIBRANT. Therefore, it appears difficult to propose general rules of phages behavior
solely based on 19 contigs representing low-quality phages genomes.

**Reply:** We applied multiple bioinformatic tools (such as VirSorter2, DeepVirfinder, VIBRANT
and CheckV, not limited to VIBRANT) to identify viral contigs as each tool can be biased due to
the different means of computing the scoring matrices. For example, VIBRANT has been found
with a lower F1 score when detecting some of the ssDNA viruses compared to VirSorter (v2)
(doi.org/10.1186/s40168-020-00990-y). The VIBRANT paper has shown some overlaps in viral
identification with the other tools (doi.org/10.1186/s40168-020-00867-0). Therefore, we applied
multiple tools and did not screen the viral contigs only based on VIBRANT results.

We clarified the benefit of the ‘voting strategy’ in the Methods: “To reduce the possibility of
introducing false positives, multiple viral bioinformatic detection algorithms were used,
incorporating stringent or suggested cut-offs.”; “Applying this voting strategy to the results from
multiple tools is intended to avoid potential limitations in one tool and minimize the detection of
false positives.”

10. - Is there a possibility that the characterized bacterial host are the most abundant organisms
of their samples? is there a correlation between the number of host and the host and/or the virus
abundancy? indeed, highly abundant sequences will be more prone to noise signal and, therefore,
to false positive detection.

**Reply:** We examined the phage hosts that were most central in our microbial network. While
some centrally located nodes did represent high abundance hosts many of the nodes also
reflected low abundance hosts. Because of this we do not believe that host abundance by itself is
the main reason for our observation that phage hosts are often the mostly high central in a
microbial network. The overlap between phage host centrality and abundance is now shown in
Table S10. The results were added and reads as the following: “We also confirmed that centrality
in the network was not merely a function of a MAG being highly abundant (Table S10). Some
MAGs that were central were abundant, but many were not, showing that abundance is not the
main driver of centrality in our host co-abundance network.”

Highly Central Nodes	Highly Abundant Contigs
drep_102	drep_7
drep_117	drep_102

drep_38	drep_79
drep_43	drep_106
drep_63	drep_5
drep_7	drep_75
drep_75	drep_15
drep_79	drep_2
drep_8	drep_3
drep_89	drep_131

Highly central nodes and highly abundant contigs are shown, overlaps between these two lists
 are in blue.

 11. - Line 168: “We calculated the average viral copies per host (VPH) to represent the degree of
 phage infection per host population.” While their calculation based on HiC links could be
 interesting, it is somehow peculiar that this value varies from 0 to 1.5 ... I would have expected a
 much higher value for lytic phages. Could the authors perform the same kind of calculations
 using Shotgun reads and compare those values?

**Reply:** Thanks for the comment. We would like to clarify that VPH represents the average virus-
 host connectivity within a given host population. For example, the value 1 could mean all
 members of the host population have 1 virus associated OR 1% of the host population has 100
 viruses. Both the viral copies per individual within a host population (V_h) and the prevalence of
 viral infection among members of a host population (F_h) determine the absolute value of VPH. F_h
 is an important factor to consider especially in systems like soil where the microbial community
 is highly diverse and the soil matrix is complex and heterogeneous. Therefore, there are chances
 that a soil sample where a larger fraction of host population (higher F_h) infected by the lysogenic
 viruses (lower count in V_h) can have a higher value of VPH. ‘a much higher value for lytic
 phages’ mentioned by the reviewer maybe not always true if F_h is considered. A lower viral
 transcriptional activity (low A_v) but a higher average VPH were detected in post-desiccation soil,
 suggesting a higher prevalence of lysogenic viral infection in the host population (higher F_h).
 This is one of the key take-homes in this study.

 We expanded the results to aid the readers in understanding this discussion. The added sentences
 in results reads as follows: “The average VPH and the VPHs of some host populations (e.g.,
 Actinobacteria) were significantly higher after soil drying ($p < 0.05$, Fig. 5a). A higher average
 VPH indicates that either a higher fraction of the host populations are infected by phages (more
 individuals infected) or a subset of the host populations are infected by a higher number of
 phages (multiple phages per individual). As such, comparing VPH alone in soils pre- and post-
 desiccation can indicate a change in phage-host relationships but cannot determine the nature of
 the change. Because we observed lower transcriptional activity and lower relative abundance for
 the phages infecting multiple hosts in post-desiccation soil, the higher VPH after soil drying may
 suggest that phage infection was more prevalent among the individuals within the host
 population (more individuals infected).”

 New sentences in the Discussion: “Therefore, we hypothesized that in desiccated soil, phages
 residing inside their hosts would be preferentially retained in a lysogenic lifecycle. In support of

this hypothesis, the host-associated phages in post-desiccation soil had lower levels of
transcriptional activity, suggestive of lysogeny. In addition, the higher average VPH in the soil
after drying indicated that a higher proportion of hosts were infected by phage, and these were
presumably lysogenic. By contrast, in pre-desiccation soil the phage infections resulted in a
significant decrease in host abundances, suggesting that the hosts were lysed by phage via the
lytic cycle. Together these results suggest that as soil dries, there is a transition of phage
lifestyles from primarily lytic to lysogenic.”

As to the point of comparing shotgun and Hi-C for the VPH values, please refer to the edits on
Line 76-80 in response to Comment 6 where we explained the two sequencing methods were
used to capture different communities and thus the values will have different meanings that are
not comparable.

Minor comments:

12. - Line 95: ref 23 is not the appropriate one to cite HiC applied to metagenomic as it is the
first HiC publication on Human

**Reply:** Thanks for the comment. We updated the citations and included the references below.

Marbouty, M., Baudry, L., Cournac, A. & Koszul, R. Scaffolding bacterial genomes and probing
host-virus interactions in gut microbiome by proximity ligation (chromosome capture) assay.

Science advances 3, e1602105 (2017).

Marbouty, M., Thierry, A., Millot, G. A. & Koszul, R. MetaHiC phage-bacteria infection
network reveals active cycling phages of the healthy human gut. Elife 10, e60608 (2021).

Reviewer #2 (Remarks to the Author):

The study showcases how Hi-C sequencing data can be used to explore virus-host linkages in
soils. According to the text, the aim of the experiment was to explore phage lifestyle comparing
wet vs dry soils. However, the experimental design included two sampling points: before and
after a 14-days incubation period. The replicates (n=3) sampled at the end were not watered (dry
soil treatment) and compared with the replicates (n=3) sampled at the beginning when water
holding capacity was 75%.

13. Ideally, a control condition made of wet soil should be considered and sampled at the end of
the experiment together with the dry soil treatment. Without that, the experiment was not
accounting for the incubation time factor. Therefore, the conclusions presented to the readers
may not be fully supported by the results observed.

**Reply:** Thanks for the comment. We agree that the time factor is an important consideration.

The text in Method was edited as the following: “The incubation of soil samples under field-
relevant conditions (i.e., 30°C for 2 weeks) mimicked the natural soil drying process in hot
weather typical of Prosser, Washington summers.”

The text in Results was edited as the following: “The Hi-C metagenome sequencing approach
was combined with shotgun metagenome and metatranscriptome sequencing to investigate how
soil drying impacts phage-host interactions. Results were compared from before and after a two-

274 week drying incubation (pre- and post-desiccation, Fig. 1) that simulated summer desiccation
that frequently occurs in these arid grassland soils.”

The text in Discussion was edited as the following: “We note that our experimental design and
data cannot address the possibility that changes occurred over time independent of the moisture
treatment, or that desiccation affected the susceptibility of hosts due to changes in outer
membrane composition.”.

14. To my knowledge this is the first study to apply Hi-C sequencing to soil. In terms of
sequencing methods, may be worth exploring the data obtained to highlight methodological
aspects and possibilities of using Hi-C sequencing for soil communities. So it may be possible to
restructure the focus of the message, to tone down the conclusions and avoid direct dry vs wet
comparisons given the experimental limitation, and instead try to emphasise the methodological
aspect of analysing Hi-C sequencing applied to soil environmental DNA. Below I provide
comments and suggestions with the intention to help to improve the study.

**Reply:** Thanks for pointing out the novel application of Hi-C in our study. As suggested by the
reviewer, we highlighted the novelty of the Hi-C sequencing approach and restructured the
Introduction, Results and Discussion sections accordingly. Changes are highlighted in blue in the
revised manuscript.

15. Would be helpful to improve Fig S1 to make more clear the experimental and computational
steps used, including how each dataset (metagenome, metatranscriptome and Hi-C) was used in
the study. Maybe this figure could be moved to main text.

**Reply:** Thanks for the suggestion. In addition to the previous Fig. S1 where we demonstrated the
experimental set-up and dataset generated, we added workflow schematics to show how the
shotgun metagenomes, Hi-C metagenomes and metatranscriptomes were co-analyzed as the
panel b. The figure is now moved to the main text as suggested (new Fig. 1).

16. It is mentioned about computational methods currently used to predict virus-host linkages
(L51-54). Why not applying those to the data to help the community providing evidence on how
much virus-host assignments can be improved with Hi-C approach? Let's say for example that by
running these computational predictions for the virus-host linkages detected with Hi-C you get
no host assigned to most (or few) of these viruses(?). Including this kind of analysis will help to
emphasise the point mentioned in the Introduction, and exploring ideas like that may help to find
new focus for the message.

**Reply:** Thanks for the suggestion. We have edited the text to emphasize that the unique value of
Hi-C is the ability to capture phage infections at the time of sampling by chemically linking
phages that enter a host cell. We also did the spacer analysis as suggested by the reviewer and
added to the results (Line 89-98) and discussion (Line 200-230).

The added results read as follows: “To demonstrate that Hi-C sequencing can capture viral
infections at the time of sampling, we compared the phage-host links detected by Hi-C with
those predicted by CRISPR spacer matching, which is currently the main bioinformatic method

for viral host prediction^{15,16}. A total of 124 CRISPR spacers recalled from the CRISPR arrays
in MAGs were matched to phage contigs, generating 121 unique phage-host links (Table S7).
Although the number of phage-host links predicted by the CRISPR spacer method and detected
by the Hi-C method are comparable, none of the Hi-C links were detected using the CRISPR
spacer approach (Tables S5 and S7). Because the CRISPR-Cas system provides an adaptive
immunity to host cells, the immunity memory based on prior viral infections¹⁷ may not detect
more recent or current viral infections. Hi-C sequencing overcomes this challenge by directly
capturing phage-host interactions at the time of sampling.”

The revised paragraph in the discussion (Line 200-230) reads as follows: “This is the first study
to provide empirical evidence of phage-host associations in soil by applying Hi-C metagenomics
to chemically link phages to their infected hosts. Previous attempts to identify hosts of soil
viruses have been challenging due to the complexity of the soil environment, coupled with the
vast diversity of largely uncharacterized soil viruses^{2,22}. As a result, the current state-of-the-
science has been to bioinformatically predict soil viral hosts from metagenomes^{5-7,16}. For
example, CRISPR spacer matching is one of the most widely used bioinformatic methods for
phage host prediction in the absence of direct evidence^{15,16}. This approach relies on the storage
of immunity memories to prior viral infections in spacer arrays in the host DNA¹⁷. The spacers
can be conserved for years in prokaryotic genomes as exact matches to the protospacer targets
from past infections²³. As a result, the number of spacers that the prokaryotic hosts acquire and
maintain can range from 10 to 100 and the majority of the spacers acquired do not represent
current viral infections²⁴. This may partly explain why there were no overlaps in phage-host
links predicted by the CRISPR matching method and those detected by Hi-C sequencing.
Therefore, although CRISPR matching and other current bioinformatic approaches are valuable
for revealing potential phage-host interactions^{5-7,16}, they are deficient in the identification of
specific hosts that phage are infecting at the time of sampling. The Hi-C approach provides an
important advance for soil microbial ecology by chemically linking phages to their bacterial host
cells and generating experimental evidence of infection at the time of sampling. We
acknowledge, however, that both of these approaches are subject to under-sampling considering
the high diversity of soil phages². With deeper sequencing, we anticipate that there would be
some overlap between phages detected using the two approaches. Currently, the combination of
both methods is complementary and provides better coverage of soil phages than using only one
approach. We also acknowledge that the Hi-C approach does not detect all possible phage-host
pairs in complex soil samples because it relies on the extraction of host cells and their associated
phage from the bulk soil matrix prior to sequencing. This could be a reason for the lower relative
abundance of Hi-C-detected host-associated phages in this soil study (5.3% to 15.0%) compared
to that in other systems that have used this approach to date; such as the human gut (~94%,
assigned to MAG)¹⁴ and wastewater treatment plants (~37%)²⁵. The lower proportion of
infected hosts in soil implies that a large proportion of the bacterial hosts were not extracted from
the soil matrix. Alternatively, a higher fraction of the soil phage community consists of free
phages that have not infected their hosts. Regardless of these shortcomings, the findings reported
here emphasize the value of the Hi-C approach for identifying ‘real-time’ phage-host pairs in soil
for the first time. These findings also revealed the impacts of changes in soil conditions on phage
infections of the soil microbiome.”

17. The metagenomes were obtained from 0.25g of soil while the Hi-C sequencing was obtained
from suspensions of 5g of soil using water. So discrepancies are expected when comparing
diversity of viruses and host between metagenomes and Hi-C sequencing. They can arise mainly
i) because of the difference in the amount of soil sampled (0.25g vs 5g), and ii) because only
extractable viruses and host could be captured by Hi-C sequencing. Would be important to
discuss this limitation.

**Reply:** We mentioned in the Methods (Line 375) that the Hi-C extraction protocol was
optimized for low biomass soil samples by PHASE GENOMICS. In addition, the Hi-C
sequencing reads were only mapped to the de-novo assemblies of the shotgun metagenome (Line
407-409). Although Hi-C may assign more viruses to hosts in the 5g of soil, we only include the
virus-host links that were detected in the viral sequences identified in the shotgun metagenomes
sequenced from the 0.25g of soil. Therefore, the impact of the amount of soil should not be a
major driver in our results.

We added sentences to the results and discussion acknowledging the second point. The added
sentences read as follow: “In contrast to shotgun metagenomics, Hi-C metagenomics was able to
identify the subpopulation of the total viral community that entered and infected host cells that
were extracted from the soil.’ ‘We also acknowledge that the Hi-C approach does not detect all
possible phage-host pairs in complex soil samples because it relies on the extraction of host cells
and their associated phage from the bulk soil matrix prior to sequencing.”

18. How the viral contigs from Hi-C sequencing data were identified? Was it in the same way as
the contigs from the metagenomes? MAGs from metagenomes were not recovered?

**Reply:** We re-arranged and added more details to the Method section (Line 391-430) and added
a workflow diagram to Fig. 1 to address the question in the revised manuscript.

To briefly explain here, Hi-C sequencing only recorded which reads were linked. The Hi-C reads
were mapped to the de-novo assemblies of the shotgun metagenome. The shotgun assemblies
were subjected to viral contig identification. When the linked Hi-C reads were mapped to one
viral contig and one bacterial contig, a virus-host pair was identified. Viral contigs were not
identified by Hi-C sequencing but by the shotgun metagenome. Because of the linkage
information from Hi-C sequencing, the MAGs obtained from shotgun metagenome can be
refined.

19. How was lysogeny inferred?

**Reply:** The lysogeny in post-desiccation soil was inferred by 1) lower transcriptional profiles of
the host-associated viruses and 2) non-lytic impact on host populations relative to what was
observed in pre-desiccation soils. This information was synthesized in the following sentences
on Line 236-241, “Therefore, we hypothesized that in desiccated soil, phages residing inside
their hosts would be preferentially retained in a lysogenic lifecycle. In support of this hypothesis,

the host-associated phages in post-desiccation soil had lower levels of transcriptional activity,
suggestive of lysogeny. In addition, the higher average VPH in the soil after drying indicated that
a higher proportion of hosts were infected by phage, and these were presumably lysogenic. By
contrast, in pre-desiccation soil the phage infections resulted in a significant decrease in host
abundances, suggesting that the hosts were lysed by phage via the lytic cycle. Together these
results suggest that as soil dries, there is a transition of phage lifestyles from primarily lytic to
lysogenic.”

20. The manuscript might benefit of using the virocell concept.

**Reply:** Thanks for the suggestion. We also considered this and recognize that the concept of the
virocell also emphasizes the fate of the host cell after being infected by viruses
(doi.org/10.1038/ismej.2012.110). However, our study does not have relevant data to define the
metabolic status of each infected host cell and may introduce more uncertainties if we use the
virocell concept. These are our concerns but we are also open to more reviewer suggestions for
better incorporation of this concept in this manuscript.

21. L24: (and elsewhere): Is it 100% safe to generalise using the term 'infection' based only on
Hi-C? Would be important to discuss that.

**Reply:** Since Hi-C captured viruses that had entered the host cells via the chemical links
between the viral and host DNAs, those host cells by definition were infected by the detected
viruses. ‘infection’ has been used in previous studies where the Hi-C was applied to capture the
virus-host links in the human gut (doi.org/10.7554/eLife.60608). ‘infect’ was used in another Hi-
C application to a rumen study ([doi: 10.1186/s13059-019-1760-x](https://doi.org/10.1186/s13059-019-1760-x)). To avoid confusion, we
added a sentence in the text (Line 77-80) to define infection in this manuscript. The sentences
read as the following: “In contrast to shotgun metagenomics, Hi-C metagenomics was able to
identify the subpopulation of the total viral community that entered and infected host cells
extracted from the soil. Specific phage-host interactions were captured using the Hi-C approach
through chemical cross-linking of the phage and host DNA molecules that were co-localized
within the same cell at the time of sampling (Fig. 1).”

22. L367: Provide more details about the method used to recover MAGs.

**Reply:** We added more information as suggested on Line 419-423: “Briefly, Hi-C mappings
were filtered with a minimum score of 20 after removing the Hi-C metagenomic paired-end
reads that were non-uniquely mapped to contigs or mapped to the same contig (length of the
contig > 1 kb). The filtered Hi-C mappings were then used to generate a contig-contig interaction
network. These contigs were further clustered into genome bins using a proprietary Markov
chain Monte Carlo (MCMC)-based algorithm⁶⁰. The genome bins or MAGs were assessed for
quality using CheckM (v1.2.0)⁶⁰.”

23. L379: Do you mean 'min' score of 0.5?

**Reply:** Yes, we now corrected it.

24. L429-430: Wouldn't it be easier for the reader if V and H were simply called as vOTU and
host abundances, respectively?

**Reply:** Thank you for the suggestion. We edited it accordingly (Line 480-482).

25. L462: clarify here that abundance refers to MAG activity/transcript?

**Reply:** The reviewer is correct that transcript data was used to construct the co-occurrence
network. We chose to use the transcript because we had metatranscriptomic data from a wider
range and number of samples than metagenomic data. We chose to use the single-copy
housekeeping gene *argS* as a proxy because it was present in a majority of the MAGs. We edited
the text (Line 521-524) to clarify this as follows, "Because we had metatranscriptomic data from
a wider range and number of samples than metagenomic data, the microbial co-occurrence
networks were constructed from metatranscriptomic data to maximize the number of input
samples. The *argS* gene was selected as the proxy because it is a housekeeping gene that is
single-copy in most organisms and was present in a majority of the MAGs."

26. L388: please clarify the tools used for the clustering step (was it the same as in L396?)

**Reply:** It is not a bioinformatic tool but part of the viral workflow that was previously published.
We added the GitHub link to the documentation of the clustering method in addition to the
citation (Line 435-436).

27. L391: provide more details on how the vOTU proteomic tree was constructed.

**Reply:** We added more information on how the tool works. The added sentences are the
following: "In brief, the pairwise similarities of the vOTU representative sequences were
computed based on the tBLASTx results. The distance between the viral sequences was
demonstrated in a dendrogram that was further clustered using the neighbor-joining algorithm."
More details can be found in the ViPTree citation specified.

28. Figure 4: consider converting back to natural numbers as they are more biologically
informative than log values. Have you also tried to correlate the activity/transcripts?

**Reply:** Thank you for the comment. VPH values were log-transformed in panels b and c to make
it easier for readers to compare the different regression results in pre- and post-desiccation soils.

We agree that the natural numbers are also informative, so we kept the natural numbers of VPH
values detected in each unique host taxon in panel a as mentioned by the reviewer.
If we understand the second comment correctly, the reviewer suggests correlating the VPH with
the host activity/transcript. The number of host bin transcripts could be hard to interpret here
because different metabolic and/or replication activities of the same host can respond differently
pre- and post-soil desiccation. Given the datasets we have, the host activity assessment is out of
the scope of this study. We acknowledge the value of this correlation suggested by the reviewer
that would be interesting to investigate in future study or in a less complex or tractable
microbiome. It is not possible to do this type of analysis on viral activity because, as we have
demonstrated, some viruses infect multiple hosts, and thus there is not the one-to-one
relationship required for regression analysis.

REVIEWER COMMENTS

Reviewer #1 (Remarks to the Author):

The revised version of the manuscript by Wu et al., is a better and clearer version. I thank the authors to answer many of my comments/questions. The different parts of the results sections are easier to understand, especially about the VPH ratio. The overall discussion is also more nuanced and open to different interpretations.

I still have two comments concerning the publication:

1- It still remains extremely difficult to have a direct and global overview of the quality of the HiC data. All the tables are full of data and difficult to understand. I re-ask my previous question as it is really important to have these data in order to have a global overview of the quality: "Very few statistics are provided concerning the data (size of the assemblies, mapping back rate, proximity ligation). It appears extremely difficult to evaluate the quality of their HiC data (raw reads, mapping rate, 3D ratio, noise signal). Indeed, no matrices are provided, and it is impossible to evaluate the noise signal encompass in their HiC data. Authors should, at least, provide a matrix of the different MAGs obtained as in the following publication. (<https://journals.asm.org/doi/10.1128/MRA.01523-19>)."

Could the authors provide a simple table with the 3D ratio for each HiC library and a contact map encompassing the different reconstructed MAGs as a supplementary Figures ?

authors can have a look here : <https://journals.asm.org/doi/10.1128/MRA.01523-19>

2- Line 77: "Hi-C metagenomics was able to identify the subpopulation of the total viral community that entered and infected host cells extracted from the soil." This affirmation needs to be demonstrated. Indeed, several HiC related publications on metagenomes and cited by the authors have emphasized the fact that HiC could also capture free viral particles. Authors should lower this affirmation or remove this sentence.

Martial Marbouty

Reviewer #2 (Remarks to the Author):

Authors have carefully addressed the comments, clarified the limitations and improved the initial version. Their study will contribute to future research applications of Hi-C to track virus-host interactions in soils. (obs: the new Fig S1 needs more information in the legend - only two points for shotgun pre-desiccation?)

REVIEWER COMMENTS

Reviewer #1 (Remarks to the Author):

The revised version of the manuscript by Wu et al., is a better and clearer version. I thank the authors to answer many of my comments/questions. The different parts of the results sections are easier to understand, especially about the VPH ratio. The overall discussion is also more nuanced and open to different interpretations.

I still have two comments concerning the publication:

1- It still remains extremely difficult to have a direct and global overview of the quality of the HiC data. All the tables are full of data and difficult to understand. I re-ask my previous question as it is really important to have these data in order to have a global overview of the quality: "Very few statistics are provided concerning the data (size of the assemblies, mapping back rate, proximity ligation). It appears extremely difficult to evaluate the quality of their HiC data (raw reads, mapping rate, 3D ratio, noise signal). Indeed, no matrices are provided, and it is impossible to evaluate the noise signal encompass in their HiC data. Authors should, at least, provide a matrix of the different MAGs obtained as in the following publication. (<https://journals.asm.org/doi/10.1128/MRA.01523-19>)."

Could the authors provide a simple table with the 3D ratio for each HiC library and a contact map encompassing the different reconstructed MAGs as a supplementary Figures ?

authors can have a look here : <https://journals.asm.org/doi/10.1128/MRA.01523-19>

2- Line 77: "Hi-C metagenomics was able to identify the subpopulation of the total viral community that entered and infected host cells extracted from the soil." This affirmation needs to be demonstrated. Indeed, several HiC related publications on metagenomes and cited by the authors have emphasized the fact that HiC could also capture free viral particles. Authors should lower this affirmation or remove this sentence.

Martial Marbouty

Reviewer #2 (Remarks to the Author):

Authors have carefully addressed the comments, clarified the limitations and improved the initial version. Their study will contribute to future research applications of Hi-C to track virus-host interactions in soils. (obs: the new Fig S1 needs more information in the legend - only two points for shotgun pre-desiccation?)

REVIEWER COMMENTS

Reviewer #1 (Remarks to the Author):

The revised version of the manuscript by Wu et al., is a better and clearer version. I thank the authors to answer many of my comments/questions. The different parts of the results sections are easier to understand, especially about the VPH ratio. The overall discussion is also more nuanced and open to different interpretations.

Response: Thanks for your positive feedback.

I still have two comments concerning the publication:

1- It still remains extremely difficult to have a direct and global overview of the quality of the HiC data. All the tables are full of data and difficult to understand. I re-ask my previous question as it is really important to have these data in order to have a global overview of the quality: "Very few statistics are provided concerning the data (size of the assemblies, mapping back rate, proximity ligation). It appears extremely difficult to evaluate the quality of their HiC data (raw reads, mapping rate, 3D ratio, noise signal). Indeed, no matrices are provided, and it is impossible to evaluate the noise signal encompass in their HiC data. Authors should, at least, provide a matrix of the different MAGs obtained as in the following publication. (<https://journals.asm.org/doi/10.1128/MRA.01523-19>)."

Could the authors provide a simple table with the 3D ratio for each HiC library and a contact map encompassing the different reconstructed MAGs as a supplementary Figures ?

authors can have a look here : <https://journals.asm.org/doi/10.1128/MRA.01523-19>

Response: We summarized the quality matrix/index suggested by the reviewer and added them to the new Supplementary Table 1 (Table S1). Quality indexes per data package/datatype were included in each sheet of the excel file (Supplementary Table 1A-E).

- The '3D ratio' per Hi-C metagenome library is in Column F of Supplementary Table 1C; We referred to the definition of '3D ratio' specified in Marbouty, Martial, et al. (2021) [<https://doi.org/10.7554/eLife.60608>] and generated the ratio using the same criteria.
- The other related statistics for each Hi-C library can be found in Columns G to J of Supplementary Table 1C, including the fraction of read pairs from the same strand, the fraction of duplicate reads, fractions of informative read pairs (MAPQ > 0, not PCR duplicates, and map to different contigs or >10 Kbp apart; cells L51-62), the fraction of intercontig read pairs.
- The number of quality reads in each shotgun and Hi-C metagenome is in Column E of Supplementary Table 1A and Column E of Supplementary Table 1C;
- The number of total and viral contigs (or 'size of the assemblies') and N50 (if applied) per co-assembled shotgun metagenomes in Rows 5-7 of Supplementary Table 1B;
- The number of bins and related statistics per sample is in Rows 11-16 of Supplementary Table 1B.

The contact map per sample was generated and included in Supplementary Figure 2 as suggested.

2- Line 77: "Hi-C metagenomics was able to identify the subpopulation of the total viral community that entered and infected host cells extracted from the soil." This affirmation needs to be demonstrated. Indeed, several HiC related publications on metagenomes and cited by the authors have emphasized the fact that HiC could also capture free viral particles. Authors should lower this affirmation or remove this

sentence.

Response: Thanks for the comment. The sentence was deleted as suggested.

Reviewer #2 (Remarks to the Author):

Authors have carefully addressed the comments, clarified the limitations and improved the initial version. Their study will contribute to future research applications of Hi-C to track virus-host interactions in soils. (obs: the new Fig S1 needs more information in the legend - only two points for shotgun pre-desiccation?)

Response: Thanks for the positive comment.

Two of the shotgun pre-desiccation data points overlapped in the NMDS plot. Instead, we now demonstrated the sample clustering in dendrograms. More details of the plots were added to the legend as suggested.

REVIEWERS' COMMENTS

Reviewer #1 (Remarks to the Author):

I have no more comments concerning the manuscript and i thanks the authors to have provided the different asked data.

Martial Marbouty